# POLAR: Policy-based Layerwise Reinforcement Learning for Stealthy Backdoor Attacks in Federated Learning

## Abstract

Federated Learning (FL) enables decentralized model training across multiple clients without exposing local data, but its distributed feature makes it vulnerable to backdoor attacks. Despite early FL backdoor attacks modifying entire models, recent studies have explored the concept of backdoor-critical (BC) layers, which poison the chosen influential layers to maintain stealthiness while achieving high effectiveness. However, existing BC layers approaches rely on rule-based selection without consideration of the interrelations between layers, making them ineffective and prone to detection by advanced defenses. In this paper, we propose **POLAR** (**PO**licy-based **LA**yerwise **R**einforcement learning), the first pipeline to creatively adopt RL to solve the BC layer selection problem in layer-wise backdoor attack. Different from other commonly used RL paradigm, POLAR is lightweight with Bernoulli sampling. POLAR dynamically learns an attack strategy, optimizing layer selection using policy gradient updates based on backdoor success rate (BSR) improvements. To ensure stealthiness, we introduce a regularization constraint that limits the number of modified layers by penalizing large attack footprints. Extensive experiments demonstrate that POLAR outperforms the latest attack methods by up to 40% against six state-of-the-art (SOTA) defenses.

## 1 Introduction

Federated Learning (FL) facilitates distributed model training across clients without centralizing private data. This paradigm has proved transformative in areas like healthcare (Ahmed et al., 2025), pharmaceuticals (Hanser et al., 2025), and finance (Aljunaid et al., 2025), where legal or privacy concerns prohibit the centralization of sensitive information. However, FL's decentralized structure also opens the door to *backdoor attacks*—stealthy modifications introduced by an attacker who injects a hidden trigger. Model would yield targeted misclassifications when encountered with the trigger, while appearing normal on non-triggered inputs (Bagdasaryan et al., 2020; Bhagoji et al., 2019; Yang et al., 2024).

Yet, two major challenges arise in backdoor attack on FL: **1) Tradeoff between Stealthiness & Effectiveness.** Stealthy attacks rely on subtle changes, whereas effective attacks require stronger perturbations; insufficient changes weaken the backdoor, while excessive ones increase detectability by robust defenses (Yang et al., 2024). **2) Generalizability.** Attacks tuned to one model or distribution often fail to transfer due to shifting structural and perturbation characteristics. Achieving robustness under dynamic FL environments thus constraining the generalizability.

Backdoor attack methods on FL can be broadly categorized by the extent of parameter modifications: **model-wise attacks**, which perturb the entire model's parameters to embed malicious behavior, and **layer-wise attacks**, which selectively modify specific layers for stealthier and more targeted manipulation. Model-wise attacks show advantages at generalizability over different models and data distributions, thus most of the existing backdoor attack methods focus on model-wise attacks, such as BadNets (Gu et al., 2019), DBA (Xie et al., 2020) and AGR (Yang et al., 2024). But model-wise attacks have poor tradeoff between stealthiness and effectiveness. Their attack surface on a model-wise level triggers large parameter changes that robust defenses can easily detect.

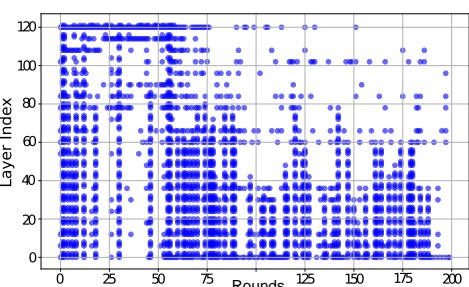

Figure 1: Selected layer distribution of LP Attack.

To address the tradeoff between stealthiness and effectiveness, a promising strategy is to target only *backdoor-critical (BC) layers* (Zhuang et al., 2024; Choe et al., 2025), a subclass of layer-wise attacks. By modifying just a small set of these influential layers, attackers can substantially alter outputs while keeping malicious updates less conspicuous, as smaller changes generally reduce the risk of detection. Nevertheless, this tradeoff remains unsatisfactory. LP Attack (Zhuang et al., 2024), for example, often fails against recent defenses. As shown in Figure 1, LP Attack tends to modify many different layers rather than consistently focusing on BC layers, leading to unstable layer selection. Moreover, its limited improvements come at the expense of generalizability, performing poorly on compact architectures such as CNNs. These shortcomings arise from its rule-based BC layer selection via forward–backward substitutions, which overlook inter-layer dependencies. As a result, LP Attack frequently makes excessive layer modifications in certain rounds, increasing detectability and ultimately degrading effectiveness.

These limitations highlight the need for a better optimization strategy for BC layer selection. However, since layers are discrete hyperparameters without gradient access, standard optimization techniques are inapplicable. The most thorough approach

| Attack | Stealthiness | Effectiveness | Generalizability | Time |
|---|---|---|---|---|
| ALL (model) | Low | Medium | High | Low |
| LPA | Medium | Medium | Medium | Medium |
| POLAR | High | High | High | Medium |
| Enum. | High | High | Low | Very High |

Table 1: Comparison of layer-selection attacks.

is brute-force enumeration, testing all possible layer combinations to identify the most effective strategy. Yet, this becomes more computationally intractable as the number of layers grows, with complexity $O(2^N)$ for a model with $N$ layers. A degraded alternative is to modify all layers, which effectively becomes a model-wise attack with an overly large and detectable attack surface. Therefore, a more adaptive and efficient approach is needed. We summarize the performance of four layer-selection strategies in Table 1, with detailed results in the Appendix B.3.

To address these issues, we proposed **POLAR** (**PO**licy-based **LA**yerwise **R**einforcement learning), a novel backdoor attack framework for FL. Since model layers are discrete hyperparameters without gradient flow, making layer selection hard to optimize. Reinforcement Learning (RL) is well-suited for this setting, as its policy design allows for flexible, global training over layer selection. Unlike prior RL-based attacks (Zhou et al., 2025; Li et al., 2023) that incur high computational overhead, which is inapplicable in FL scenarios. POLAR makes the RL process lightweight by using Bernoulli sampling to explore the action space efficiently. POLAR applies policy-gradient optimization to dynamically select layers to poison across FL rounds, capturing inter-layer dependencies. This adaptive design enables exploration over the full action space and convergence to optimal and robust layer-selection strategies, offering strong generalizability and scalability. Additionally, we designed a regularization term to guide POLAR to focus on the BC layers more efficiently, thus balancing stealthiness and effectiveness. As summarized in Table 1, POLAR achieves high stealthiness, effectiveness, and generalizability with moderate runtime. Extensive experiments show that POLAR outperforms the latest attack methods by up to 40% against six state-of-the-art (SOTA) defenses across various models and datasets.

To sum up, our contributions are as follows:

- We propose POLAR, which is the first framework to leverage RL for solving the layer-selection problem in layer-wise backdoor attacks. By formulating layer selection as an RL task, POLAR adaptively identifies the most effective group of BC layers to poison. Its learning-based design further provides flexibility and resilience, enabling the attack to adapt under diverse defenses in the FL process.
- POLAR finds a great balance between stealthiness and performance by leveraging real-time feedback on backdoor success rate (BSR) from the aggregation process, and constraints on the layer selection. Experiments validate that POLAR outperforms SOTA attack methods in most settings on BSR, main task accuracy and malicious client acceptance rate (MAR) under SOTA defenses.
- POLAR shows great generalizability for its learning-based feature, making it available for different models and datasets, even under restrained attack scenarios. Moreover, by changing the

BSR-based reward function, POLAR can be applied to other FL settings. The scalability makes POLAR practical in real FL deployments.

## 2 RELATED WORK

### 2.1 BACKDOOR ATTACKS IN FEDERATED LEARNING

**Model-wise Backdoor Attacks.** BadNets (Gu et al., 2019) introduces fixed triggers and target labels into training data, creating strong trigger-target associations. However, its centralized assumption and reliance on large-scale data poisoning make it impractical for FL, where data is decentralized. Its global perturbations are also easily detected by robust defenses. DBA (Xie et al., 2020) distributes trigger patterns across clients, allowing covert backdoor injection, but still modifies global parameters, making it vulnerable to statistical anomaly detection. AGR (Yang et al., 2024) amplifies adversarial updates by adaptively reweighting gradients, but its heuristic weighting overlooks structural dependencies across layers, leaving it less effective against strong defenses.

**Layer-wise Backdoor Attacks.** To enhance stealthiness, recent methods (Fang et al., 2020; Li et al., 2023) selectively manipulate layers, inspired by findings that small parameter changes can yield large effects (Stich, 2019; Rothchild et al., 2020; Li et al., 2017). LP Attack (Zhuang et al., 2024) targets specific *backdoor-critical* layers to reduce detectability, while SDBA (Choe et al., 2025) improves stealthiness by masking gradients and periodically refreshing poisoned layers, but relies on heuristic rules without modeling inter-round dynamics. However, both methods use rule-based selection and fail to model inter-round dependencies, limiting their adaptability under evolving defenses.

### 2.2 DEFENSES IN FEDERATED LEARNING

To guard against backdoor attacks, various robust aggregation defenses have been proposed. Some methods cluster and filter out anomalous updates, such as FLAME (Nguyen et al., 2022), FLARE (Wang et al., 2022). Others use robust statistics or trust scores, such as MultiKrum (Blanchard et al., 2017), FLTrust (Cao et al., 2021), RLR (Ozdayi et al., 2021), and some detect deviations via gradient-pattern analysis, like FLDetector (Zhang et al., 2022).

### 2.3 REINFORCEMENT LEARNING'S APPLICATION IN FEDERATED LEARNING

In practical FL settings, client data is often **non-IID**, posing challenges for anomaly detection, as traditional aggregation assumes homogeneous updates. Reinforcement learning (RL), especially policy gradient methods (Williams, 1992; Sutton et al., 1999), offers a natural fit for FL by enabling adaptive strategies based on feedback from each round. Prior works like FAVOR (Wang et al., 2020) use RL to optimize aggregation under non-IID settings.

Being inspired, from an adversarial perspective, aggregator feedback can similarly guide RL-based backdoor attacks. However, existing approaches (Zhou et al., 2025; Li et al., 2023) often suffer from high computational overhead, limiting their practicality in FL.

For brevity, additional details on these related works are provided in the Appendix C.

## 3 THREAT MODEL

**Attacker's Objective.** The attacker aims to inject a backdoor into the global model so that inputs embedded with a predefined trigger are misclassifed into a target class $y_{\text{target}}$, while maintaining high performance on benign inputs. Additionally, the aggregation would be finished using FedAvg (McMahan et al., 2017), and also SOTA defenses.

**Attacker's Knowledge.** The attacker is assumed to know the model architecture and training protocol, have full access to and control over its own local training data. Also, the attacker is able to observe its own aggregation feedback from the central server. However, it has no access to other clients' data, model updates, or defense configuration.

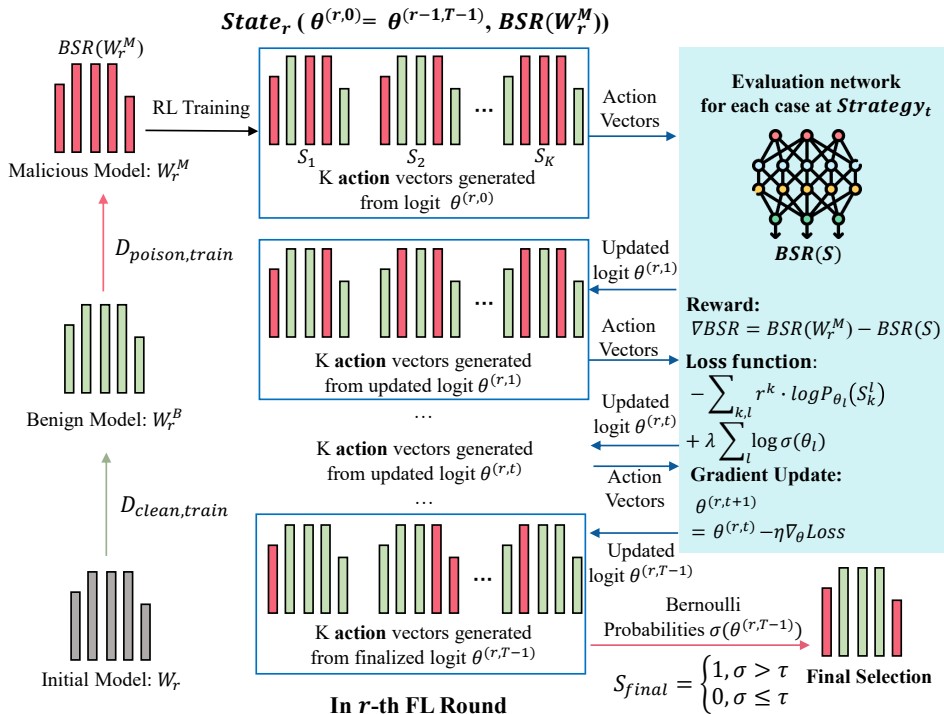

**Figure 2:** Overview of POLAR. In each FL round $r$, the adversary trains both benign and poisoned models to evaluate BSR. A RL agent maintains and updates a logit vector $\theta(r, t)$, from which $K$ binary action vectors are sampled via Bernoulli distributions to determine which layers to poison. The policy is optimized via gradient updates based on BSR rewards. After $T$ steps, the final logit is thresholded to obtain the layer selection $S_{\text{final}}$, which is used to generate the final malicious update submitted to the server.

**Attacker's Capability.** The attacker controls a small fraction of the total clients (lower than 10%) in the FL system. These compromised clients can coordinate their strategies and share information. The attacker has full access to their local data and model parameters. The attacker would receive the global model snapshot in each communication round, allowing direct manipulation of both data and weights during training.

## 4 METHODOLOGY

### 4.1 OVERVIEW OF POLAR FRAMEWORK

In each FL training round, POLAR operates as a feedback-driven module that adaptively determines the subset of model layers to inject malicious weights into, the workflow is shown in Figure 2. In the POLAR agent, after generating the malicious weights, it processes the Bernoulli samples according to an initial logit (set to 0 at the beginning), which sampling discrete distribution on a lightweight scale by reducing the redundancy in action space. It calculates and updates the current policy with MDP results (reward, loss and gradient update), which continues the loop in the RL batches. Then, POLAR selects the layers according to the policy. Finally, the malicious attacker submits the partially injected model, as shown in the right corner of Figure 2. And the strategy logit obtained from the previous RL round would work as the initial logit in the next round. These adaptively chosen layers are subsequently incorporated into the malicious update. The partially modified malicious model updates are aggregated by the server alongside benign updates, completing the attack iteration.

### 4.2 ENVIRONMENTAL CUES

By modifying REINFORCE (Williams, 1992), POLAR learns a policy that chooses layers to maximize the backdoor success rate (BSR), which serves as the reward, while minimizing the number of layers selected to remain stealthy. In essense, the POLAR agent will try different layer-selection

strategies, observe the BSR, and update its layer-selection policy to favor actions that yield higher BSR. In general, we aim to learn a layer-selection policy $D_\theta$ (a distribution over layer selections defined by the parameters $\theta$). POLAR's policy refinement is achieved through gradient ascent. We maximize the expected BSR of the modified model with selected layers $S$: $\mathbb{E}_{S \sim D_\theta}[BSR(S)]$. For batch size $K$, $N$ layers, for each sample $k(k = 1, ..., K)$, and each layer $l(l = 1, ..., N)$, let $S_k^{(l)} \in \{0, 1\}$ be the indicator of whether layer $l$ is selected (action 1) or not (action 0), together with its logit $\theta_l$ by sampling $K$ independent selections $S_1, S_2, ..., S_K$ from $D_\theta$, the optimized gradients is:

$$\nabla_\theta \mathbb{E}_{S \sim D_\theta}[BSR(S)] \approx \sum_{k=1}^{K} BSR(S_k) \sum_{l=1}^{N} \nabla_\theta \log(P_{\theta_l}(S_k^{(l)})). \tag{1}$$

To avoid selecting too many layers or no layers, POLAR only selects the most critical layers, **regularization** term is added. $p_l$ is the probability of selecting layer $l$, and $p_l = \sigma(\theta_l)$, $r_k$ is the reward (score) for sample $k$, $\lambda$ is the regularization hyperparameter. Thus, the regularization term could be written as $\lambda \sum_{l=1}^{N} log(p_l)$, and we can conclude the full **loss** function $L$ as:

$$L = - \sum_{k=1}^{K} \sum_{l=1}^{N} (S_k^{(l)} \cdot log(p_l) + (1 - S_k^{(l)}) \cdot log(1 - p_l)) \cdot r_k + \lambda \sum_{l=1}^{N} log(p_l). \tag{2}$$

This loss essentially measures the negative log-likelihood of the actions taken, weighted by how beneficial those actions turned out to be (the reward). The gradient of this loss with respect to the policy parameters (the logits) is then used to update the policy, making it more likely to select layers that yield a higher reward in future rounds.

### 4.3 ACTIONS LEARNING

POLAR learns a dynamic attack strategy where the layer selection adapts in real time based on feedback from the aggregator. By modeling layer selection as an RL policy, POLAR continuously refines its attack footprint to maximize BSR while minimizing detection risks. We frame the layer selection process in POLAR as Markov Decision Process (MDP). In this MDP, the **state** can be considered as the current global model parameters, and the **action** is the choice of a subset of layers to replace with malicious ones. After the action, the attacker receives a **reward** based on the BSR of the modified model. POLAR's learning process is refined according to the RL algorithm REINFORCE (Williams, 1992).

**Action**: The actions represent binary decisions for each layer. Specifically, for a given layer $l$, the action $S^{(l)}$ is 1 if the layer is selected for malicious replacement and 0 if the layer remains unchanged. The overall action vector for the $k$-th sample is

$$S_k = (S_k^{(1)}, S_k^{(2)}, ..., S_k^{(N)}). \tag{3}$$

**State**: At each FL round $r$, the state representation in POLAR incorporates key contextual information required by the RL agent to make informed decisions. Specifically, the state includes the final logit obtained from previous round $\theta^{r-1}$, which decides the selection cases of previous step. We set backdoor success rate (BSR) of malicious model $W_r^M$ obtained from global model $W_r$ as $BSR(W_r^M)$, evaluated BSR of the strategy obtained from previous step $BSR(S_{r-1})$. The state at round $r$ is represented as:

$$\text{State}_r = \{\theta^{r-1}, BSR(W_r^M)\}. \tag{4}$$

This comprehensive state design provides sufficient information for the RL agent to dynamically assess and optimize subsequent layer-selection actions in evolving FL environments.

**Reward**: After the agent selects an action for the batch $k$ (the action vector $S_k$ shows the layers to be replaced with malicious weights), the modified candidate model is tested to obtain its backdoor success rate $BSR(S_k)$. By comparing the candidate model's performance to a baseline malicious model (with backdoor success rate $BSR(W_r^M)$), this evaluation yields the reward signal:

$$Reward = BSR(S_k) - BSR(W_r^M). \tag{5}$$

This reward is then used to update the agent's policy via the REINFORCE algorithm.

## 4.4 COMPUTATIONAL COMPLEXITY

Let $K$ be the batch size, which is the number of samples, $T$ be the number of RL steps, $N$ be the total number of layers in the model attacked, $E$ be the cost to evaluate the BSR via poisoned model each time called the evaluation, $h$ be the cost to sample one binary action, which is negligible at $O(1)$, additionally let $l$ be the number of layer selected.

In LP Attack, for $l$ layers being chosen, in one FL local epoch, the total cost of LP Attack is: $O(N \cdot E + l)$. In POLAR, for one FL local epoch, the total cost of POLAR is: $O(K \cdot T \cdot (N + E + l))$.

Since the primary computational cost lies in the evaluation, we can approximate the total cost to $O(L \cdot E)$ for LP Attack, $O(K \cdot T \cdot E)$ for POLAR. Both methods therefore operate with linear time complexity.

More detailed proof about the algorithm design can refer to Appendix A.

## 5 EVALUATION

### 5.1 EXPERIMENTS SETTINGS

**Datasets and Models.** We conduct experiments using NVIDIA RTX 4090 and GRID A100X-10C. We evaluate the performance of attack using two widely used benchmark datasets: CIFAR10 and Fashion-MNIST. Following previous studies (Zhuang et al., 2024), we use VGG19 (Simonyan & Zisserman, 2015) and ResNet18 (He et al., 2015) for CIFAR10, simple five-layer CNN for Fashion-MNIST. We simulate a Non-IID data distribution following prior works (Cao et al., 2021), with $q = 0.5$.

**Baseline Defenses and Attacks.** We assess the backdoor attack on six state-of-the-art (SOTA) or representative defenses in FL: FLARE (Wang et al., 2022), FLDetector (Zhang et al., 2022), FLTrust (Cao et al., 2021), FLAME (Nguyen et al., 2022),RLR (Ozdayi et al., 2021), and MultiKrum (MK) (Blanchard et al., 2017). To ensure a comprehensive evaluation of attacks' effectiveness, we also evaluate the backdoor attacks under FL aggregation scenarios with FedAvg (McMahan et al., 2017) without defenses, the parameters setting for the defenses following Zhuang et al. (2024). We compare POLAR against two representative attacks in FL: BadNets (Gu et al., 2019) and LP Attack (Zhuang et al., 2024), which also serves as the baseline of POLAR.

**Metrics.** Following prior work on FL backdoor attacks (Zhuang et al., 2024; Gu et al., 2019; Xie et al., 2020), we evaluate the global model using two primary metrics: main task accuracy and backdoor success rate (BSR). Given the dynamic nature of the global model in FL, we report both the average BSR (ABSR) and best BSR (BBSR) over the last 10 communication rounds. To evaluate the stealthiness of the attack, we adopt two metrics introduced in LP Attack (Zhuang et al., 2024): the malicious client acceptance rate (MAR) and the benign client acceptance rate (BAR). MAR measures the proportion of rounds where malicious clients successfully bypassed defense and were selected for aggregation, while BAR represents the average proportion of rounds in which benign clients were selected for aggregation.

The attacker setup and more detailed experimental settings are in Appendix B.1.

### 5.2 THE EFFECTIVENESS OF POLAR

To evaluate POLAR in realistic federated settings, we test it under non-IID conditions on CIFAR-10 (ResNet18 and VGG19) and Fashion-MNIST (CNN).

As shown in Table 2, POLAR consistently achieves the BSR and main task accuracy (Acc) across most settings. Compared to LP Attack and BadNets, POLAR demonstrates clear advantages under strong defenses. BadNets fails almost entirely under modern defenses such as FLARE, FLDetector, MK, and FLAME with BSRs below 20%. LP Attack, while more stealthy, suffers from lower effectiveness especially under adpative defenses such as FLARE and RLR, and struggles to generalize across architectures like VGG 19, where POLAR outperforms it by over 15%.

Figure 3 and Figure 4 further indicate that POLAR converges faster and selects layers more effectively than LP Attack. Thus, when the communication epochs is restricted in a range, POLAR could

| Model (Dataset) | | ResNet18 (CIFAR-10) | | | VGG19 (CIFAR-10) | | | CNN (Fashion-MNIST) | | |
|---|---|---|---|---|---|---|---|---|---|---|
| Attack | | LP Attack | POLAR | BadNets | LP Attack | POLAR | BadNets | LP Attack | POLAR | BadNets |
| FedAvg | BBSR | 94.71±0.82 | **96.28**±0.17 | 94.37 | 93.43±0.93 | **95.01**±0.89 | 86.28 | 86.01±2.30 | 96.73±0.55 | **100** |
| | ABSR | 90.21±0.84 | **90.65**±1.68 | 90.93 | 89.44±2.01 | **91.37**±0.15 | 78.69 | 80.70±0.80 | 94.82±0.65 | **100** |
| | Acc | 77.62±0.52 | **77.68**±0.78 | 76.70 | 79.48±0.93 | **81.22**±1.73 | 78.33 | 88.57±0.17 | **88.60**±0.30 | 88.31 |
| FLARE | BBSR | 78±1.05 | **87.36**±1.24 | 11.88 | 90.53±2.61 | 91.0±1.08 | **96.19** | 86.06±1.03 | **96.82**±0.63 | 3.28 |
| | ABSR | 59.9±0.33 | **73.63**±1.17 | 5.79 | 65.04±2.52 | **74.65**±1.53 | 45.76 | 76.3±1.46 | **92.44**±0.13 | 2.63 |
| | Acc | 71.26±0.11 | **72.43**±0.36 | 70.89 | 68.82±0.67 | **69.85**±0.64 | 58.7 | **88.62**±0.69 | 88.60±0.47 | 88.34 |
| FLDetector | BBSR | 97.65±1.08 | **98.1**±0.66 | 8.44 | 93.57±1.11 | **96.09**±1.00 | 17.13 | 98.51±0.23 | **98.95**±0.87 | 73.68 |
| | ABSR | **97.01**±1.92 | 96.36±1.37 | 7.39 | 85.05±1.02 | **87.9**±0.95 | 16.86 | 95.99±0.16 | **97.08**±0.67 | 67.23 |
| | Acc | 60.46±1.60 | 62.53±0.36 | **65.22** | 51.04±0.23 | **58.92**±0.14 | 57.13 | 75.3±0.01 | **79.2**±0.48 | 79.13 |
| FLTrust | BBSR | 96.67±28.78 | **96.77**±7.99 | 92.67 | 81.07±32.35 | **96.08**±8.76 | 11.77 | 87.92±3.54 | **92.43**±0.37 | 73.27 |
| | ABSR | 84.3±29.19 | 88.2±2.62 | **89.43** | 62.14±28.76 | **77.51**±1.98 | 6.33 | 77.67±6.15 | **82.84**±0.55 | 69.13 |
| | Acc | 74.87±8.49 | **75.63**±0.28 | 74.89 | 74.57±2.33 | 75.86±0.81 | **76.83** | **89.1**±0.61 | 88.47±0.23 | 88.79 |
| FLAME | BBSR | **95.35**±0.78 | 94.58±0.21 | 28.08 | 86.99±3.39 | **90.18**±11.19 | 51.23 | 83.9±0.28 | **86.93**±0.53 | 0.33 |
| | ABSR | **93.03**±0.11 | 92.71±0.37 | 7.59 | 59.6±11.58 | **75.8**±2.31 | 8.33 | 75.74±0.62 | **78.56**±0.34 | 0.08 |
| | Acc | 69.71±0.23 | 71.9±0.80 | **73.47** | **63.45**±1.03 | 51.75±9.61 | 63.1 | 87.04±0.23 | 87.78±0.78 | **87.89** |
| RLR | BBSR | 87.54±1.69 | **93.3**±0.83 | 78.33 | 85.36±1.17 | **91.7**±1.47 | 78.77 | 0.44±0.56 | **25.72**±3.71 | 19.34 |
| | ABSR | 80.64±0.70 | **90.55**±0.30 | 59.78 | 67.32±0.61 | **89.23**±1.56 | 73.54 | 0.03±0.04 | 11.55±7.40 | **16.33** |
| | Acc | 72.2±0.51 | 73.77±0.26 | **75.39** | **75.4**±0.73 | 74.67±0.88 | 67.65 | 86.28±0.09 | **87.75**±3.60 | 85.79 |
| MK | BBSR | 92.38±4.09 | **95.49**±1.28 | 13.86 | 92.21±2.13 | **96.8**±1.41 | 25.2 | 91.5±1.65 | **91.91**±0.80 | 1.34 |
| | ABSR | 87.94±0.47 | **91.24**±0.95 | 3.59 | 41.48±4.24 | **83.43**±1.22 | 7.26 | **77.85**±1.42 | 76.73±0.56 | 0.09 |
| | Acc | 66.49±9.89 | 74±0.37 | **74.32** | 61.39±1.41 | **65.21**±1.34 | 58.42 | **87.48**±0.31 | 87.71±0.18 | 87.67 |

Table 2: Main task accuracy and BSR on Non-IID datasets. We mark the highest BSR and Acc as bold within the same setting. The LP Attack is LP Attack Zhuang et al. (2024). The results are the average of five repeated experiments. For LP Attack and POLAR, $a \pm b$, where $a$ is the mean value, and $b$ is the standard deviation. Acc: main task accuracy (%), BSR unit: %.

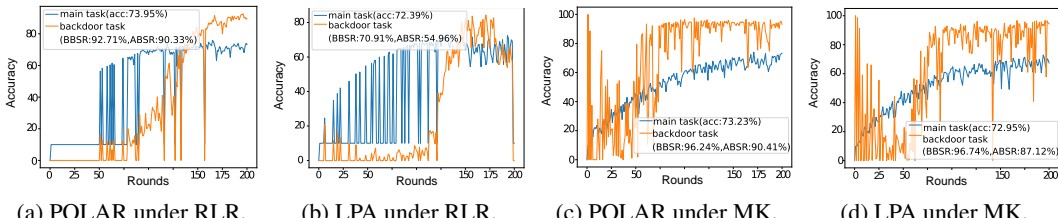

(a) POLAR under RLR.  (b) LPA under RLR.  (c) POLAR under MK.  (d) LPA under MK.

Figure 3: Comparison of POLAR and LP Attack performance under RLR and MK defenses on ResNet18.

more efficiently attack the task than LP Attack. Across every tested combination, POLAR reaches strong BSR values while preserving accuracy in non-IID conditions. POLAR's RL-based, adaptive layer selection enables high stealthiness and strong backdoor success, outperforming both heuristic (LP Attack) and static (BadNets) methods across all evaluated models and defenses.

## 5.3 THE STEALTHINESS OF POLAR

**Higher Acceptance Rate.** To evaluate stealthiness, we compare the Benign Acceptance Rate (BAR) and Malicious Acceptance Rate (MAR) of PO-LAR, BadNets (Gu et al., 2019), and LP Attack (Zhuang et al., 2024) on CIFAR-10 and Fashion-MNIST under both IID and non-IID settings. We choose MultiKrum (MK) as one of the defence following the LP

| Model (Dataset) | Attack | MK (IID) | | MK (non-IID) | | RLR (IID) | | RLR (non-IID) | |
|---|---|---|---|---|---|---|---|---|---|
| | | MAR↑ | BAR↓ | MAR↑ | BAR↓ | MAR↑ | BAR↓ | MAR↑ | BAR↓ |
| VGG19 (CIFAR10) | LP Attack | **0.98** | 0.41 | 0.39 | 0.41 | 0.48 | 0.39 | 0.86 | 0.32 |
| | POLAR | 0.97 | 0.42 | **0.49** | **0.39** | 0.98 | 0.44 | **0.97** | **0.32** |
| | BadNets | 0.01 | 0.44 | 0.13 | 0.43 | **0.98** | **0.39** | 0.82 | 0.37 |
| ResNet18 (CIFAR10) | LP Attack | 0.88 | 0.35 | 0.68 | 0.37 | 0.74 | **0.29** | 0.88 | 0.35 |
| | POLAR | **0.92** | **0.34** | **0.75** | 0.36 | 0.98 | 0.56 | **0.93** | 0.34 |
| | BadNets | 0.00 | 0.44 | 0.04 | 0.44 | 0.97 | 0.43 | 0.60 | 0.39 |
| CNN (FMNIST) | LP Attack | 0.51 | 0.39 | 0.66 | 0.37 | 0.04 | 0.11 | 0.00 | 0.70 |
| | POLAR | **1.00** | **0.33** | **0.99** | **0.34** | **0.56** | 0.14 | **0.34** | 0.56 |
| | BadNets | 0.00 | 0.44 | 0.00 | 0.44 | 0.22 | 0.17 | 0.29 | 0.66 |

Table 3: Detection accuracy of MK and RLR on CIFAR-10 and Fashion-MNIST. MAR indicates malicious client acceptance rate, and BAR indicates benign client acceptance rate.

Attack (Zhuang et al., 2024), and RLR as another for its adapative features. The results in Table 3 show that MK and RLR successfully prevent most malicious updates by BadNets attack, which also validates that RLR and MK can easily distinguish malicious and benign updates by non-layerwise attack methods during the aggregation process. The high MAR and BAR of POLAR indicate that POLAR can successfully have its malicious updates accepted by the FL servers and bypass the detection on all the settings. Meanwhile, POLAR outperforms both LP Attack and BadNets in most of the scenarios, especially when LP Attack has poor performance under MK defense. The pro-

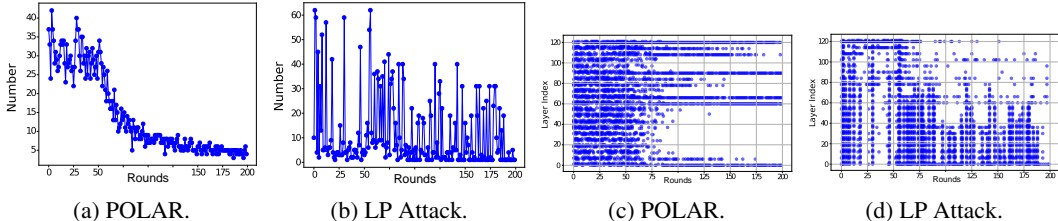

(a) POLAR.  (b) LP Attack.  (c) POLAR.  (d) LP Attack.

Figure 4: Comparison between POLAR and LP Attack. **Left:** Number of layers selected under RLR defense. **Right:** Distribution of layers selected.

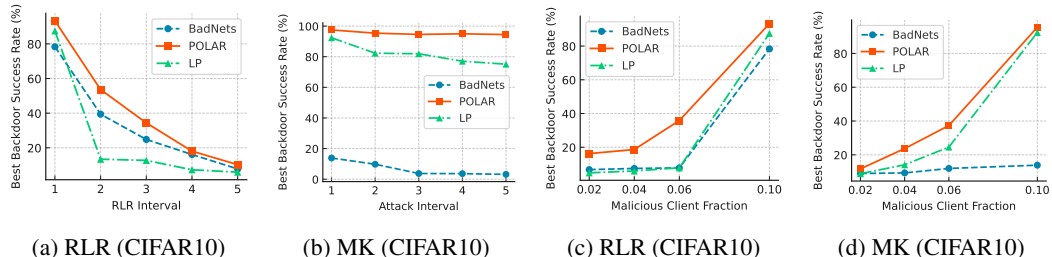

(a) RLR (CIFAR10)  (b) MK (CIFAR10)  (c) RLR (CIFAR10)  (d) MK (CIFAR10)

Figure 5: Comparison of POLAR and LP Attack on ResNet (CIFAR10). **Left:** Impacts of different attack intervals $F$ on the attack performance ($F$ indicates that an attack is performed every $F$ rounds). **Right:** Impacts of different proportions of malicious clients ($C = 0.02, 0.04, 0.06, 0.1$) on the attack performance in the fixed-pool setting.

motion validates that POLAR could raise the stealthiness by minimizing the attack surface with policy-based learning rules epoch by epoch for its unsupervised feature and adaptation to defenses.

**The Efficiency in targeting crucial layers.** Furthermore, we visualize the selection of layers by plotting a line graph and scatter graph about the layers being selected under MK defense, with non-iid scenarios, with ResNet18 training on the dataset CIFAR10, as is shown in Figure 4. We notice that POLAR can steadily decrease the number of crucial layers, while LP Attack has relatively large fluctuations during the process. Consequently, POLAR is shown to successfully minimize the attack surface with its design. Although MK is an adaptive defense, POLAR converged to attack specific layers, which validates that POLAR are more likely to find the real BC layers than LP Attack. And when the training epoch of FL is restricted, POLAR is more likely to target the crucial layers. This phenomenon helps explain POLAR's great promotion in stealthiness, as well as the steady performance under different defenses.

## 5.4 GENERALIZABILITY OF POLAR

**Impact of attack intervals.** We evaluate the performance of our attack compared to baseline attacks under different attack intervals, as shown in Figure 5. The attack interval $F$ ranges from $1$ to $5$, where $F$ indicates that the attacker participated once every $F$ rounds. The setting is designed to simulate scenarios where the frequency of malicious participation is limited, thus requiring more effective and stealthy attack strategies.

The results show that with $F$ increasing, the performance of all attacks would degrade. This trend is consistent across all evaluated attack methods, where a lower malicious participation frequency results in reduced BSR. We observe that BadNets fails in all cases. Notably, POLAR maintains a comparable BSR even when the attack interval is as high as $F = 5$ under both RLR and MK. However, LP Attack lost almost all its effectiveness under RLR, which demonstrates its sensitivity to defenses when attack opportunities are restricted. Under MK, we observe that the BSR of LP Attack drops rapidly when interval $F = 2$ to about 80%, but POLAR remains high BSR in all frequencies we set. This experiment verifies POLAR's generalizability in attack strength when malicious participation is restricted.

**Impact of different proportions of malicious clients.** To assess the generalizability of POLAR in practical settings, we evaluate its performance under a fixed-pool attack, where malicious clients are drawn from a fixed pool with varying participation ratios $C \in \{0.02, 0.04, 0.06, 0.1\}$.

| Penalty | ABSR (%) | BBSR (%) | MAR | Selected Layer |
|---------|----------|----------|------|----------------|
| 0 | 76.24 | 81.17 | 0.88 | 8 |
| 5 | 85.58 | 91.37 | 0.89 | 8 |
| 10 | **90.55** | **93.30** | 0.93 | 5 |
| 20 | 37.75 | 39.18 | **0.97** | 5 |

Table 4: Effect of different penalty weights in POLAR.

| Batch Size | ABSR (%) | BBSR (%) | MAR | Selected Layer |
|------------|----------|----------|------|----------------|
| 10 | 31.94 | 43.93 | 0.69 | 7 |
| 25 | 46.90 | 64.06 | 0.74 | 6 |
| 50 | **90.55** | **93.30** | **0.93** | 5 |

Table 5: Effect of batch size in POLAR.

In this setup, a random subset of the malicious pool is selected each round, simulating real-world conditions where malicious clients may appear randomly across communication rounds.

As shown in Figure 5, POLAR consistently achieves the highest BSR across all settings, significantly outperforming LP Attack and BadNets, even at very low malicious ratios. This demonstrates POLAR's robustness and persistence under limited adversarial presence.

Both LP Attack and POLAR show improved performance as the malicious ratio increases, owing to their adaptive learning mechanisms. In contrast, BadNets performs consistently, as it lacks an adaptive component. LP Attack's inferior performance stems from slower convergence caused by its static, rule-based layer selection, whereas POLAR maintains more effective learning even under limited participation due to its policy-driven adaptability. More experiment results on generalizability are in Appendix B.2.

## 5.5 ABLATION STUDY

To further evaluate the robustness and sensitivity of POLAR to training parameter choices, we additionally explore two key parameters: the penalty weight $\lambda \in \{0, 5, 10, 20\}$ and the sampling batch size $B \in \{10, 25, 50\}$ under the RLR defense using ResNet-18 on the CIFAR-10 dataset.

As shown in Table 4, a low or zero penalty leads POLAR to select more layers, including non-essential ones, resulting in reduced stealthiness and only moderate backdoor effectiveness. When the penalty increases, POLAR learns to focus on a smaller set of BC layers, improving stealthiness (higher MAR) while maintaining high effectiveness (ABSR and BBSR). However, an excessively large penalty overly restricts the attack, causing a significant drop in both ABSR and BBSR. These findings indicate that a moderate penalty achieves the best balance between effectiveness and stealthiness.

Table 5 reveals the impact of sampling size during layer selection. With smaller batch sizes, POLAR struggles to obtain reliable gradient signals from sampled actions, leading to instability in policy learning and degraded performance across all metrics. As the batch size increases, the policy is trained on more representative samples, improving its ability to distinguish BC layers and optimizing both effectiveness and stealthiness. Notably, with $B = 50$, POLAR achieves its best performance, indicating that adequate sampling is essential for stable and successful policy optimization in our framework.

The results confirm that POLAR is robust to moderate parameter variations and highlight that carefully tuned settings, $\lambda = 10$ and $B = 50$ in our paper, can significantly enhance its attack performance while preserving stealthiness.

More ablation study would be discussed in Appendix B.3.

## 6 CONLUSION

We propose POLAR, the first RL-based layer-wise backdoor attack in FL, which formulates layer selection as a discrete optimization problem via policy-gradient methods. By leveraging real-time feedback from the aggregation process, POLAR adaptively selects backdoor-critical layers to balance stealthiness and effectiveness. Extensive experiments across various scenarios demonstrate that POLAR outperforms existing attacks like LP Attack and BadNets in terms of backdoor success rate, main task accuracy, and malicious client acceptance rate under state-of-the-art FL defenses. Furthermore, POLAR demonstrates strong generalizability, maintaining effectiveness across architectures, defense strategies and attack settings. Visualization results confirm its ability to efficiently identify minimal yet impactful layer subsets, making it both robust and scalable in practical FL deployments.

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

# APPENDIX

## A   EXTENDED METHODOLOGY DISCUSSION

### A.1   PROOF OF DESIGN

By modifying REINFORCE (Williams, 1992), we mainly refine the gradient in our algorithm design. In general, we aim to learn a layer-selection policy $D_\theta$ (a distribution over layer selections defined by the parameters $\theta$). POLAR's policy refinement is achieved through gradient ascent. We maximize the expected BSR of the modified model with selected layers $S$: $\mathbb{E}_{S \sim D_\theta}[BSR(S)]$. For batch size $K$, $N$ layers, for each sample $k(k = 1, ..., K)$, and each layer $l(l = 1, ..., N)$. By sampling $K$ independent selections $S_1, S_2, ..., S_K$ from $D_\theta$:

$$\nabla_\theta[\mathbb{E}_{S \sim D_\theta}[BSR(S)]] \approx \sum_{k=1}^{K} \nabla_\theta P_\theta(S_k) BSR(S_k)$$

$$\approx \sum_{k=1}^{K} P_\theta(S_k) BSR(S_k) \nabla_\theta \log(P_\theta(S_k)), \tag{6}$$

where for a model with $N$ layers, the logit of layer $l$ is denoted as $\theta_l$. For the layer selection set $S \in \{0, 1\}^N$, where each element represents the selection of a layer, the probability of $S$ is $P(S)$. For each layer $l$, the probability of being chosen is $P(S^{(l)}) \in [0, 1]$, and $k$-th selection case is $S_k$. The second approximation is achieved by log-derivative trick:

$$\nabla_\theta P_\theta(S_k) = P_\theta(S_k) \nabla_\theta \log(P_\theta(S_k)). \tag{7}$$

Also, we have the probability of selecting each layer given by the Bernoulli distribution:

$$P_{\theta_l}(S^{(l)}) = \sigma(\theta_l)^{S^{(i)}} (1 - \sigma(\theta_l))^{1 - S^{(l)}}, \tag{8}$$

where $\sigma(\cdot)$ is the sigmoid function. Since the choices of each layer are independent, borrowing the idea from factorizing the decision-making process to simplify the optimization problem (Zoph & Le, 2017; Pham et al., 2018; Li et al., 2017), we have

$$P_\theta(S) = \prod_{l=1}^{N} P_{\theta_l}(S^{(l)}). \tag{9}$$

Hence, we have the final expression of our optimized gradient:

$$\nabla_\theta \mathbb{E}_{S \sim D_\theta}[BSR(S)]$$

$$\approx \sum_{k=1}^{K} BSR(S_k) \sum_{l=1}^{N} \nabla_\theta \log(P_{\theta_l}(S_k^{(l)})). \tag{10}$$

For each sample in the batch and for every layer, the loss accumulates terms of form $-log(P_{\theta_l}(S_l)) \cdot r$, where $r$ is the corresponding reward, $p$ is the probability of certain layer being chosen. The policy gradient loss is initially computed according to different action taken:

$$L = \begin{cases} -\log(1 - p) \cdot r & \text{if action = 0 (Layer not selected).} \\ -\log(p) \cdot r & \text{if action = 1 (Layer selected).} \end{cases} \tag{11}$$

Thus, the complete **loss** function $L$ can be summarized as follows:

$$L = -\sum_{k=1}^{K} \sum_{l=1}^{N} (S_k^{(l)} \cdot log(p_l)$$

$$+ (1 - S_k^{(l)}) \cdot log(1 - p_l)) \cdot r_k$$

$$+ \lambda \sum_{l=1}^{N} log(p_l), \tag{12}$$

which is consistent with the main body of the paper.

---

**Algorithm 1:** POLAR Layer Selection

---

**Input:** Global model $W_t$, Malicious model $W_M$, BSR evaluation $BSR(\cdot)$, $N$ layers , $K$ batch size, rounds $T$, learning rate $\eta$, regularization parameter $\lambda$, threshold $\tau$

**Output:** Layer selection $S_{\text{final}}$

Initialize logits $\theta_l \leftarrow 0.5, \; l = 1..N$;

**for** $t = 1$ **to** $T$ **do**

    Sample $K$ selection cases $S^{(k)} \sim \text{Bernoulli}(\sigma(\theta))$;

    Compute reward $r^{(k)} = BSR(S^{(b)}) - BSR(W^M)$;

    $\mathcal{L} \leftarrow -\sum_{k,l} r^{(k)} \cdot \log P_{\theta_l}(S_l^{(k)}) + \lambda \sum_l \log \sigma(\theta_l)$;

    $\theta \leftarrow \theta - \eta \cdot \nabla_\theta \mathcal{L}$;

$S_{\text{final}}[l] \leftarrow \mathbb{1}[\sigma(\theta_l) > \tau]$;

**return** $S_{\text{final}}$

---

**Algorithm 2:** Federated Learning with POLAR Attack

---

**Input:** Initial global model $W^0$; client datasets $\{D_i\}$; communication rounds $R$

**Output:** Final global model $W^R$

**for** $r = 0$ **to** $R - 1$ **do**

    Initialize update buffer: $\mathcal{U} \leftarrow []$;

    // Benign clients perform standard local training

    **for** *each benign client $i$* **do**

        $W_i \leftarrow \textbf{LocalTrain}(W^r, D_i)$;

        $\mathcal{U}.\text{append}(W_i - W^r)$;

    // Malicious clients perform adaptive backdoor attack

    **for** *each malicious client $j$* **do**

        $W_b \leftarrow \textbf{LocalTrain}(W^r, D_j^{\text{clean}})$;

        $W_m \leftarrow \textbf{LocalTrain}(W_b, D_j^{\text{poisoned}})$;

        $BSR \leftarrow \textbf{Evaluate}(W_m)$;

        $S \leftarrow \textbf{POLAR-RL}(W_b, W_m, BSR)$;

        $W_c \leftarrow \textbf{ReplaceLayers}(W_b, W_m, S)$;

        $\mathcal{U}.\text{append}(W_c - W^r)$;

    // Server aggregates all updates

    $W^{r+1} \leftarrow \textbf{Aggregate}(\mathcal{U})$;

**return** $W^R$

---

### A.2 PSEUDO CODE

Algorithm 1 shows the workflow of POLAR in detail. In the RL training, we can generate layer selection cases by $\theta^{(t)}$ as a strategy, so we note it as $Strategy_t$ in Figure 2 to better show the overall workflow of POLAR. In Algorithm 2, we show the FL process with the POLAR attack, which validates how POLAR is processed in the overall workflow.

### A.3 DISCUSSION ABOUT ALGORITHM DESIGN

**Why Bernoulli Sampling?** The layer-wise selection space is discrete and binary: for $l$-th layer $S^{(l)} \in \{0, 1\}$, indicating whether it is substituted or not, in our introduction part, we have already analyzed that enumerating all $2^L$ combinations is infeasible, therefore, the RL agent uses a Bernoulli sampling over a logit-based parameterization: $S^{(l)} \sim \text{Bernoulli}(\sigma(\theta_l))$, with $P_{\theta_l}(S^l) = \sigma(\theta_l)$, which is the policy network's output probability for layer $l$. This greatly reduces the redundancy in action space, which reduces memory overhead, and maintains the completeness of the policy network. Bernoulli sampling makes the RL process lightweight and scalable, which is quite suitable for FL backdoor attack.

**How to make reasonable selection?** The design of selection penalty takes both BSR and the number of selected layers into consideration, which helps POLAR avoid large and suspicious gradient deviations, a key factor in resisting robust federated defenses. It also prevents the POLAR agent from selecting no layers at all. POLAR thus introduces a novel and unprecedented RL-based layer-wise backdoor attack, and stealthiness control under an actor-critic-like RL mechanism, making it distinctly robust and flexible across a wide range of FL settings.

**Computational Complexity** Let $K$ be the batch size, which is the number of samples, $T$ be the number of RL steps, $N$ be the total number of layers in the model attacked, $E$ be the cost to evaluate the BSR via poisoned model each time called the evaluation, $h$ be the cost to sample one binary action, which is negligible at $O(1)$, additionally let $l$ be the number of layer selected.

In LP Attack, it costs $O(N)$ to substitute each layer in the original malicious model with benign weights one by one at first; and cost $O(E)$ each time the substitution happens, which costs $O(N \cdot E)$ in total, and finally choose the $l$ layers according to the threshold and finish the substitution, which costs $O(l)$; thus for one FL local epoch, the total cost of LP Attack is: $O(N \cdot E + l)$.

In POLAR, for Bernoulli sampling, it takes $O(N)$ to sample binary masks for all layers in a sample; and $O(l)$ for parameter swapping from benign one to malicious one, which is also negligible; it takes $O(E)$ to finish the model evaluation, which is the dominant cost, where forward passing over all validation data; thus for all the above repeated for $T$ RL steps, thus for one FL local epoch, the total cost of POLAR is: $O(K \cdot T \cdot (N + E + l))$.

Since the primary computational cost lies in the evaluation, we can approximate the total cost to $O(L \cdot E)$ for LP Attack, $O(K \cdot T \cdot E)$ for POLAR. Both methods therefore operate with linear time complexity.

# B SUPPLEMENTARY EXPERIMENTS

## B.1 EXPERIMENTS SETTINGS

**Attacker setup.** By default, each round selects $n = 10$ clients from $N = 100$ total clients, with $C = 10\%$ malicious. Client weights are decided by FedAvg or defense-aware strategies. The FL training runs for 200 communication rounds to make converge in most cases with two local epochs per client and a learning rate of 0.1 for both CIFAR10 and Fashion-MNIST. The trigger is a $5 \times 5$ pixel square located at the bottom-right corner of the images. For POLAR, we set $lr = 0.01, \lambda = 10$, batch number $T = 10$, and batch size $K = 50$, where POLAR can generally reach performance with both robustness and efficiency. For LP Attack, we set threshold $\tau = 0.95$, following (Zhuang et al., 2024).

**Defense Methods and Settings**

- FLARE (Wang et al., 2022): We reuse the setting in the original paper, where the root compromises 10 samples per class.

- FLDetector (Zhang et al., 2022): We reuse the setting in the original paper: Every client performs a single step of standard gradient descent and submits its corresponding model updates to the server in each round. As a result, the fraction of clients participating in the aggregation is set to $C = 1$, the local epoch is set to $E = 1$, and the number of training rounds is enlarged to $R = 500$. Additionally, the window size is set to 10 and attacks start after the server finishes the initialization.

- FLTrust (Cao et al., 2021): We enlarge the size of root dataset from 100 in the original paper to 300, which enables the server to detect attacks more accurately.

- RLR (Ozdayi et al., 2021):We set the threshold of learning rate flipping in each parameter to 4, following the original paper, where RLR claims that the threshold should be larger than the number of malicious clients (about one malicious client each round in our work).

- FLAME (Nguyen et al., 2022): The minimal cluster size is set to $n/2 + 1$, minimal sample number to 1, and the noise parameter to 0.001 following the original paper.

- MK (MultiKrum) (Blanchard et al., 2017): The server calculates the squared distance called Krum distance through the closest $N \times C - f$ clients updates, where $f$ is a hyperparameter

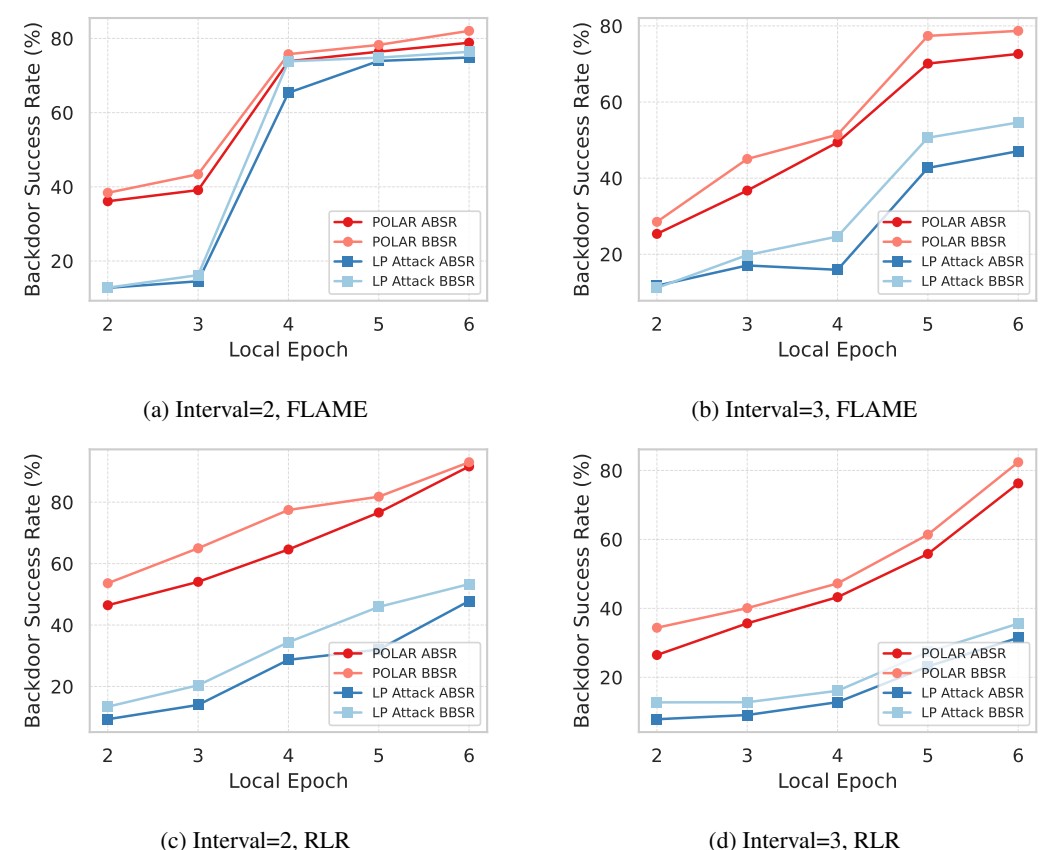

(a) Interval=2, FLAME

(b) Interval=3, FLAME

(c) Interval=2, RLR

(d) Interval=3, RLR

Figure 6: Impact of local epochs on the performance of layer selection attack with attack interval ($F = 2, 3$) under FLAME and RLR defense.

and assumed to be equal or larger than the number of malicious clients, which is 1 in our setting. In our experiments, four clients with the highest distance score are selected by MultiKrum to aggregate the global model with $f = 2$.

## B.2 GENERALIZABILITY

**Impact of local epochs for layer selection attack.** To evaluate the performance of POLAR in more realistic scenarios, we vary the local training epochs $E$ of malicious clients from 2 to 6. This aligns with our threat model, which assumes that the attacker may bypass the central server's constraints to perform additinal local training and thereby enhance attack strength. Since we have already observed stable and convergent results when the attack frequency is set to $F = 1$, we now evaluate POLAR and LPA under RLR and FLAME defenses with reduced attack frequencies $F = 2$ and $F = 3$. Notably, both methods exhibit a significant performance boost at $E = 4, F = 2$ for FLAME defense. Similarly, at $E = 5, F = 3$, both attacks have performance boost, again indicating that increased local computation can offset less frequent attack. Under RLR defense, both LP Attack and POLAR have gradually effectiveness improvement. This is because the doubled local training epochs compensates for the reduced attack opportunity, allowing both attacks to achieve BSR approaching similar effect nearly matching the performance under continuous attack ($F = 1$).

While the overall trends of POLAR and LPA are consistent, POLAR always outperforms LPA, and demonstrates a more stable attack pattern. This confirms POLAR's strength and its reduced dependency on aggressive local training to maintain high backdoor efficacy.

| Attack | RLR (IID) | | RLR (non-IID) | | MK (IID) | | MK (non-IID) | | Running Time (s) |
|---|---|---|---|---|---|---|---|---|---|
| | MAR | ABSR | MAR | ABSR | MAR | ABSR | MAR | ABSR | |
| ALL (model) | 0.88 | 88.29 | 0.78 | 87.32 | 0.79 | **98.14** | 0.47 | **97.21** | 131.68 |
| LPA | 0.74 | 65.5 | 0.88 | 80.64 | 0.86 | 89.34 | 0.68 | 87.94 | 287.89 |
| POLAR | 0.975 | 91.51 | 0.93 | 90.55 | **0.92** | 95.87 | 0.69 | 91.24 | 326.53 |
| Enum. | **0.995** | **93.94** | **0.965** | **91.16** | **0.92** | 94.54 | **0.76** | 87.29 | 789.04 |

Table 6: Initial Experiments on ResNet-18 with CIFAR-10 dataset. MAR indicates malicious clients acceptance rate, ABSR indicates average backdoor success rate (%), and results are averaged over three runs.

| Metric | BadNets (E=2) | POLAR (T=10, E=2) | LP Attack (E=2) | POLAR (T=2, E=2) | POLAR (T=5, E=2) | POLAR (T=2, E=4) | POLAR (T=5, E=4) | POLAR (T=2, E=6) | POLAR (T=5, E=6) |
|---|---|---|---|---|---|---|---|---|---|
| Time (s) | 58.58 | 126.53 | 87.89 | 54.20 | 78.31 | 107.93 | 161.59 | 140.64 | 197.53 |
| ABSR | 59.78 | **90.55** | 80.64 | 66.16 | 85.65 | 82.00 | 89.09 | 88.89 | 89.22 |
| BBSR | 78.33 | 93.30 | 87.54 | 88.64 | 92.63 | 92.14 | **95.40** | 93.09 | 93.07 |
| MAR | 0.59 | **0.975** | 0.86 | 0.64 | 0.935 | 0.895 | **0.975** | 0.965 | **0.975** |

Table 7: Ablation study on POLAR against RLR defense with different parameter settings. $T$: RL steps per episode; $E$: malicious local epochs. Best results are in **bold**, second best are underlined.

## B.3 ABLATION STUDY

**Comparison between different layer selection attack methods.** To validate the claim in Table 1 as introduced in the Introduction, we conduct experiments on ResNet-18 with the CIFAR-10 dataset under both RLR and MK defenses. Due to the exponential time complexity of the full search space ($O(2^N)$), the Enumeration method is limited to evaluating all two-layer combinations. As shown in Table 6, POLAR consistently achieves a strong trade-off between stealthiness and effectiveness, while maintaining a reasonable runtime. Although Enumeration yields the highest MAR, it incurs a significant computational cost of 789.04 seconds. Meanwhile, if we directly attack all layers, which makes the layer-wise attack downgrade to model-wise, it can raise the effectiveness compared to LPA at the risk of lower stealthiness. In conclusion, POLAR demonstrates superior practicality by achieving near-optimal performance with far lower overhead, making it a more scalable and stealthy solution for layer-wise backdoor attacks.

**Sensitivity to different parameters in FL training.** Based on our analysis of the theoretical computational costs of POLAR and LP Attack, we observe that POLAR's runtime is primarily influenced by the number of RL training steps. To explore this trade-off, we conduct an ablation study by changing the training parameters to explore lightweight versions of POLAR, and examine whether increased local training can compensate for the reduced batch count.

As shown in Table 7, lowering the training steps significantly reduces runtime. However, when the local epoch is fixed at $E = 2$, performance deteriorates notably with training steps $T = 2$. Even at $T = 5$, the performance remains suboptimal. In contrast, as the number of local training epochs increases, the performance of all downgraded versions of POLAR improves, eventually matching or surpassing the original POLAR while maintaining perfect stealthiness with high MAR. These results demonstrate POLAR's adaptability to varying attack settings and its ability to maintain effectiveness under computational constraints.

## C EXTENDED RELATED WORK

### C.1 FEDERATED LEARNING

Federated learning (FL) trains a global model by aggregating updates from $n$ distributed clients, each with a local dataset $D_i$. Let $f_i(\cdot)$ denote the local empirical loss and $p^{(i)} = \frac{|D_i|}{\sum_j |D_j|}$ be the relative data weight. At each communication round $t$, the server selects a subset of clients $\mathcal{N}_t$ and sends them the global model $W_t$. Each client $i \in \mathcal{N}_t$ initializes its local model as $W_{t+1}^{(i)} \leftarrow W_t$ and performs multiple steps of gradient descent on $f_i$, yielding an updated model: $W_{t+1}^{(i)} = W_t - \beta \nabla f_i(W_t; D_i)$. The server aggregates all received models using FedAvg (McMahan et al., 2017) or a defense-aware strategy to obtain $W_{t+1}$.

## C.2 Backdoor Attacks in Federated Learning

**Model-wise Attack.** BadNets (Gu et al., 2019) injects a fixed trigger into training data and assigns it a target label, inducing a strong trigger-target association for reliable misclassification. However, it assumes centralized control and requires large-scale data poisoning, which is impractical in FL due to decentralized training and limited access to other clients' data. Moreover, its global parameter perturbation is easily detected by robust FL defenses.

Distributed Backdoor Attack (DBA) (Xie et al., 2020) distributes trigger patterns across compromised clients, making each local update appear benign. This allows the global model to learn the backdoor covertly. However, its model-wide parameter changes still introduce detectable statistical anomalies under strong FL defenses.

Adversarial Gradient Reweighting (AGR) (Yang et al., 2024) enhances backdoor attacks in FL by adaptively reweighting client gradients during aggregation, amplifying adversarial contributions while suppressing benign ones. This design improves stealthiness and effectiveness without directly modifying model parameters and integrates naturally into the FL pipeline. However, AGR relies on heuristic gradient magnitudes, overlooking structural and inter-layer dependencies, which leads to unstable performance under strong defenses and poor generalization on compact or heterogeneous models.

**Layer-wise Attack.** To evade robust defenses, recent approaches (Fang et al., 2020; Li et al., 2023) shift toward adaptive strategies that selectively manipulate model components. Studies in pruning and adversarial updates suggest that modifying a few parameters can induce large effects (Stich, 2019; Rothchild et al., 2020; Li et al., 2017).

Building on this, recent works (Zhuang et al., 2024; Choe et al., 2025) identify *backdoor-critical (BC)* layers—where limited but targeted modifications can sustain high attack success with minimal footprint. LP Attack (Zhuang et al., 2024) poisons only a few BC layers, reducing update detectability against layer-wise anomaly defenses. SDBA (Choe et al., 2025) further maintains stealthiness via static or heuristic layer selection. However, both LP Attack and SDBA assume fixed, independent layer choices, ignoring dynamic inter-layer dependencies across FL rounds. This limits their adaptability under evolving defenses.

## C.3 Defenses in Federated Learning

FL defenses aim to ensure robust and secure model aggregation in the presence of malicious clients. **FLAME** (Nguyen et al., 2022) defends backdoor attacks by clustering client updates and discarding suspicious clusters exhibiting significant deviation from benign model updates. Similarly, **MultiKrum (MK)** (Blanchard et al., 2017) enhances Byzantine robustness by aggregating only the client updates closest in parameter space, effectively filtering updates suspected to be compromised. **FLTrust** (Cao et al., 2021) leverages a trusted root dataset at the server to establish trustworthiness of client updates through cosine similarity, enabling selective integration of credible updates. Additionally, **RLR** (Ozdayi et al., 2021) identifies suspicious clients by monitoring abnormal shifts in learning rates, mitigating their influence through adaptive thresholding. **FLARE** (Wang et al., 2022) clusters updates and introduces adaptive noise to dilute malicious contributions, enhancing aggregation resilience against subtle parameter manipulations. **FLDetector** (Zhang et al., 2022) utilizes gradient-pattern analysis, identifying anomalies based on gradient norms and historical update distributions. Collectively, these defenses serve as comprehensive benchmarks to evaluate the stealthiness and effectiveness of federated backdoor attacks.

## C.4 Reinforcement Learning in FL Optimization and Attack

In realistic deployments, the data on each client is often **non-IID**, meaning the distributions underlying each dataset $D_i$ differ significantly. However, traditional Federated Learning (FL) aggregation mechanisms assume relatively homogeneous client updates, making anomaly detection more challenging. Such a challenge further underscores the complexity and importance of tackling non-IID data distributions.

Reinforcement learning (RL), particularly policy gradient methods (Williams, 1992; Sutton et al., 1999), allows an agent to refine its layer-selection strategy based on round-by-round feedback, which perfectly matches with Federated Learning (FL)'s feature, is expected to serve as a solution to non-IID scenarios. Thus, optimizing FL using RL methods has gained attention for years. FAVOR (Wang et al., 2020) leverages RL to optimize FL aggregation strategies by dynamically adjusting learning rates or participation rules, showing great performance under non-IID. Following that, more and more adaptive frameworks are proposed to enhance the aggregation and mitigation of FL, such as FLARE (Wang et al., 2022) and RLR (Ozdayi et al., 2021).

Being inspired, from the perspective of attackers, the aggregator's acceptance or rejection of model updates can serve as a natural feedback signal. More backdoor attack methods are designed based on RL. However, previous RL-based backdoor attacks on FL (Zhou et al., 2025; Li et al., 2023) risk over-high computational cost, which are not applicable under FL scenarios.

## D  EXAMPLE CODE

Example code can be found in supplementary materials. To illustrate the core idea of POLAR, we provide a simplified standalone PyTorch example simulating a layer-wise backdoor attack on a lightweight ResNet model trained over the CIFAR-10 dataset. The example consists of three main stages:

- **Benign Training:** The model is first trained for several epochs on clean CIFAR-10 data to simulate a typical federated learning client update. This reflects realistic local training in FL.
- **Layer-wise Attack Injection:** A Bernoulli mask is sampled to emulate POLAR's policy-based layer selection mechanism. Only selected layers are perturbed using small additive noise to simulate stealthy, targeted parameter manipulation. This illustrates POLAR's design principle of minimizing attack footprint by modifying only backdoor-critical layers.
- **Evaluation:** The backdoor success rate (BSR) is computed by injecting a predefined trigger (a $5 \times 5$ white square) into test images and measuring the model's classification rate into the target class. Clean accuracy is also reported before and after the attack to highlight stealthiness.

This example shows full reinforcement learning logic and federated server aggregation for simplicity. It aims to provide an interpretable and minimal working example of POLAR's key insight: adaptive, sparse, and effective layer-wise backdoor injection, which illustrates the overall logic. Due to the lack of real policy training and limited attack-time optimization, its effectiveness in this example is intentionally reduced. Real training results of POLAR under RLR defense with ResNet18 training on CIFAR-10 dataset can be found in the log file included. README file is also included for the implementation of the example code.

## E  THE USE OF LARGE LANGUAGE MODELS

In this paper, we used ChatGPT as a general-purpose writing assistant to polish grammar, improve clarity, and refine formatting. The model did not contribute to research ideation, technical content, or experimental results.

