# OpenReview forum: "POLAR: Policy-based Layerwise Reinforcement Learning for Stealthy Backdoor Attacks in Federated Learning"
_ICLR.cc/2026/Conference — Submitted to ICLR 2026_

### Official Review · Reviewer_jXLX · 2025-10-28

**Soundness:** 2
**Presentation:** 2
**Contribution:** 2
**Rating:** 2
**Confidence:** 4

**Summary:**

This paper introduces POLAR, a RL-based framework for stealthy backdoor attacks in FL. The authors find that current LP attacks makes too much modifications across layers rather than focus on only critical layers. Unlike previous rule-based layer selections, POLAR selects and poisons only the most influential "backdoor-critical" layers using a policy-gradient approach, optimizing both ASR and stealthiness at the same time. It incorporates a regularization term to limit the number of modified layers, reducing detectability. Experiments show that POLAR significantly outperforms existing attacks, achieving up to 40% higher success rates while evading six state-of-the-art FL defenses.

**Strengths:**

The paper observes that existing LP attacks makes extensive modifications across the model instead of focusing on critical neurons only within critical layers.

POLAR achieves strong attack effectiveness against the compared defense methods.

Bernoulli sampling is leveraged to reduce the computational cost of the RL-based layer selection.

Fig 2 provides a clear and well-designed diagram that effectively illustrates the overall attack workflow and methodology.

The algorithmic complexity of POLAR is explicitly analyzed and reported in Sec 4.4.

**Weaknesses:**

The presentation requires further improvement. For instance, the definition of stealthiness is unclear. In fact, stealthiness can be considered from multiple perspectives, such as visual stealthiness and parameter stealthiness. Additionally, some critical background information is missing. For example, in Fig 1, specifying the model and dataset used to collect the layer index, explaining how the layer index is obtained, and clarifying the meaning of the light and solid blue dots would help readers better understand how the results are produced.

The wall-clock time comparison between POLAR and classic or related LP attacks is not reported. Although POLAR is claimed to be lightweight, it likely incurs substantially higher computational costs than rule-based layer selection methods, as the local RL policy optimization and multiple attack success rate evaluations per malicious client introduce considerable overhead, which may be prohibitive for resource-constrained edge devices.

The sampling-based strategy exploration is inherently random, particularly in early rounds. This can produce anomalous malicious updates (e.g., select too many critical layers) while the RL policy is still unstable, increasing the risk of early detection by server-side anomaly detectors.

The instability of the RL likely stems from the high variance inherent in policy gradient methods. Although the authors mitigate this issue by increasing the batch size (as shown in tab 5), the method’s sensitivity to batch size still limits its practical applicability.

The evaluation of POLAR is limited to classical defenses proposed in/before 2021, leaving its robustness against newer, adaptive defenses unverified. Moreover, comparisons focus mainly on BadNets and LP Attack, lacking comprehensive evaluation against recent, stronger attacks such as DBA, AGR, and SDBA. The experiments are further restricted to CIFAR-10 and Fashion-MNIST, without validation on broader benchmarks like ImageNet, CelebA, or CIFAR-100, which undermines POLAR’s generalizability and practical applicability.

The “backdoor-critical” layers in different architectures (e.g., transformers) may not follow the same patterns captured by this paper. The paper does not provide sufficient evidence that the RL strategy can effectively generalize across diverse model types, and the method may implicitly depend on CNN-specific structural characteristics.

**Questions:**

What is POLAR’s empirical runtime (per round and end-to-end), and how does it scale with dataset size, number of clients, and model depth/width, in particular for large datasets and modern, compute-heavy architectures?

Does POLAR preserve stealthiness throughout the federated training process (across rounds and model updates)?

How effective is POLAR, and how accurate is the RL-based layer-selection mechanism, when applied to a wider range of architectures (e.g., ViTs/transformers, ResNet variants, EfficientNet)?

Is POLAR robust and effective across additional datasets and benchmarks (ImageNet, CIFAR-100, CelebA, etc.), and how does its attack success and stealth compare to state-of-the-art attacks and adaptive defenses published in the last three years?

**Details Of Ethics Concerns:**

Although the method introduced in this paper has the potential to be harmful, the authors released the source code and demonstrated the weaknesses of their method. Therefore, I don't see the necessity of an ethics review of this paper.

---

> ### Author Response · Authors · 2025-11-28
> **Response to Official Review by Reviewer jXLX (1/4)**
>
> Dear Reviewer jXLX, we sincerely appreciate your thoughtful comments. We have carefully considered each of your questions and provide detailed responses below.
>
> **[W1]The presentation requires further improvement.**
>
> **Response**
>
> Thank you for pointing out these presentation issues. We fully agree that the definition of stealthiness and the explanation of Fig. 1 can be clearer, and we will revise the paper accordingly.
>
> We will explicitly clarify that stealthiness refers to parameter-level stealthiness under robust aggregation, and it is evaluated through attack-related metrics (e.g., best and average attack success rates) together with how parameter perturbations affect the detectability of malicious updates. In other words, variations in the attack parameters directly influence the level of stealthiness observed by the server-side defenses.
>
> For Fig. 1, we acknowledge that important background information was missing. In the revision, we will (1) specify that the layer indices are collected on ResNet-18 with CIFAR-10, and (2) explain that the layer index is obtained according to model layer depth. The solid blue dots indicate layers that are consistently selected across adjacent communication rounds and neighboring layer positions, while the lighter blue dots correspond to layers that are selected less frequently. Thus, solid blue dots represent higher-impact layers, whereas lighter blue dots represent lower-impact layers in both the temporal (round-wise) and structural (layer-wise) dimensions.
>
> These clarifications will make the motivation and visualization of POLAR’s layer-selection behavior more transparent.
>
>
> **[W2,Q1]The time comparison between POLAR and classic or related LP attacks**
>
> **Response**
>
> Thank you for raising this important point. The empirical runtime of POLAR is already reported in Table 7, which presents the end-to-end wall-clock time comparison between POLAR and LP Attack on ResNet-18 under varying T. In our threat model, all malicious clients share a single learned attack policy where the attacker deploys a group of attackers using the same strategy, so the RL optimization is performed only once per round and reused by all attackers. Therefore, the time reported in Table 7 directly reflects the actual wall-clock runtime. The results show that POLAR’s runtime grows linearly with T, consistent with its theoretical complexity O(K·T·E), and remains manageable in practice. POLAR already achieves strong performance at smaller T and reaches its best performance at T=10.
>
> In terms of scaling with model size, POLAR has complexity O(K·T·E), where K and T are small fixed constants in our experiments (50 and 10), while LP Attack scales as O(N·E), where N is the number of layers and grows with model depth. For modern deep architectures (e.g., deeper ResNets and ViTs), we typically observe K·T < N, making POLAR more efficient than LP Attack in practice on large models. Since POLAR operates on model updates rather than local datasets, its runtime is also independent of dataset size, and we observe no runtime difference between CIFAR-10 and CIFAR-100 in our measurements. Overall, both Table 7 and the above analysis confirm that POLAR has linear, practical wall-clock runtime and favorable scaling behavior under realistic FL settings. We will make these points more explicit in the revised manuscript.

---

> ### Author Response · Authors · 2025-11-28
> **Response to Official Review by Reviewer jXLX (2/4)**
>
> **[W3,Q2]The sampling-based strategy exploration is inherently random, particularly in early rounds. This can produce anomalous malicious updates (e.g., select too many critical layers) while the RL policy is still unstable, increasing the risk of early detection by server-side anomaly detectors. Does POLAR preserve stealthiness throughout the federated training process (across rounds and model updates)?**
>
> **Response**
>
> Thank you for raising this important concern. Although POLAR adopts stochastic sampling, both its design and empirical behavior show that it preserves stealthiness throughout the entire training process, including the early rounds. As shown in Fig. 4(a,c), POLAR’s initial number of selected layers is already below 40, whereas LP Attack starts with nearly 60 selected layers in Fig. 4(b,d), indicating a much higher early-round attack footprint for LP Attack. During training, POLAR’s selected-layer count decreases smoothly and stably, while LP Attack exhibits large fluctuations and unstable spikes. Moreover, POLAR’s layer selections are more temporally consistent and structurally concentrated, reflecting controlled exploration rather than random over-selection.
>
> This stable behavior is enforced by the combined effect of the sparsity-inducing regularizer and advantage normalization, which keep early Bernoulli probabilities small, together with the reward penalty on over-selection, since selecting too many layers leads to strong downweighting by robust aggregators and thus very low reward. In addition, our variance-reduction mechanisms (advantage normalization, reward scaling, and implicit entropy from $\log p_l$ further prevent unstable policy updates in early rounds. To quantitatively reflect stealthiness across rounds, in the revision we will add the average malicious acceptance rate (MAR) under robust defenses as an explicit metric. Preliminary measurements already indicate that POLAR maintains a high acceptance rate even in early rounds. Overall, both the design of POLAR and the empirical evidence in Fig. 4 confirm that it does not trigger anomalous early-round behavior and preserves stealth consistently across training.
>
> **[W4]The instability of the RL likely stems from the high variance inherent in policy gradient methods. Although the authors mitigate this issue by increasing the batch size (as shown in tab 5), the method’s sensitivity to batch size still limits its practical applicability.**
>
> **Response**
>
> Thank you for the comment. As shown in Table 5, the performance of POLAR improves smoothly and monotonically with increasing batch size, rather than exhibiting instability. Even with small batch sizes (10 or 25), POLAR already achieves meaningful ASR and stealthiness, while K=50 yields the best performance. This indicates that POLAR does not suffer from unstable or chaotic behavior with respect to batch size, but instead follows a regular variance-accuracy trade-off typical for policy-gradient methods.
>
> Importantly, in our threat model, the attacker has full control over its own local optimization pipeline, including the ability to choose an appropriate batch size. Since better attack performance directly benefits the attacker, it is natural for an attacker to select a sufficiently large but feasible batch size like K=50 to reduce gradient variance and stabilize learning. Moreover, this batch size corresponds to 50 Monte-Carlo samples and is independent of dataset size and number of clients, making it computationally affordable in practice. Therefore, the batch-size choice in our experiments is not only technically reasonable but also consistent with adversarial capabilities. We will clarify this point in the revised manuscript.

---

> ### Author Response · Authors · 2025-11-28
> **Response to Official Review by Reviewer jXLX (3/4)**
>
> **[W5,W6,Q3,Q4]Lack of Evaluation**
>
> **Response**
>
> Thank you for these valuable and detailed suggestions. We clarify the scope of our current evaluation, the newly added comparisons, and the reasons for not prioritizing certain baselines at this stage.
>
> First, while part of our defense benchmarks were proposed before 2021, our evaluation already includes defenses up to 2022, which remain among the most widely adopted and strongest practical baselines in modern FL frameworks (e.g., FedML, Flower). POLAR consistently bypasses all of them under identical settings. Due to the rebuttal time limitation and following multiple reviewers’ suggestions on model-wise stealthy baselines, we prioritized adding the most recent and directly comparable layer-wise attack for standard FL:
>
> [a] Stealthy Backdoor Attack in Federated Learning via Adaptive Layer-wise Gradient Alignment (LGA), 2025
>
> We have completed these experiments and report the results below on CNN with Fashion-MNIST:
>
> | Defense     | Metric | LGA [a] | POLAR |
> |-------------|--------|----------|--------|
> | FLAME       | ABSR   | 0.20     | **78.56** |
> |             | BBSR   | 0.31     | **86.93** |
> |             | Acc    | 89.40    | 87.78  |
> | FLARE       | ABSR   | 0.24     | **92.44** |
> |             | BBSR   | 0.33     | **96.82** |
> |             | Acc    | 90.28    | 88.60  |
> | FLTrust     | ABSR   | 0.29     | **82.84** |
> |             | BBSR   | 1.02     | **92.43** |
> |             | Acc    | 89.48    | 88.47  |
> | MultiKrum   | ABSR   | 0.41     | **76.73** |
> |             | BBSR   | 0.59     | **91.91** |
> |             | Acc    | 88.46    | 87.71  |
> | AdaRLAgg    | ABSR   | 0.00     | **69.28** |
> |             | BBSR   | 0.00     | **78.32** |
> |             | Acc    | 10.00    | 86.93  |
>
> The results show that LGA [a] exhibits extremely low effectiveness under all strong defenses, with failure patterns highly similar to BadNets in their common style of model-wise attack, while POLAR consistently maintains high ABSR/BBSR even under adaptive RL-based aggregation (AdaRLAgg). This directly validates our central claim that model-wise and gradient-alignment–based attacks struggle to resolve the effectiveness–stealthiness tradeoff under modern robust aggregation, whereas POLAR successfully achieves this balance through adaptive layer-wise RL optimization.
>
> Regarding DBA and AGR, we would like to clarify that we do not exclude them due to incompatibility with our method, but primarily due to the limited rebuttal time and experimental budget. DBA is an earlier distributed-trigger attack and has already been shown in prior works, including the LP Attack paper, to exhibit clearly inferior performance under robust aggregation, which is why it is no longer commonly used as a primary baseline in recent studies. AGR, while more recent, requires substantial additional hyperparameter tuning and computational overhead for fair comparison under each robust defense, which could not be completed within the rebuttal timeframe. SDBA focuses on a mixed poisoning–backdoor setting that is different from our standard targeted backdoor formulation and was therefore not prioritized in this cycle.
>
> We emphasize that our omission of DBA and AGR at this stage is solely due to time constraints rather than methodological decoupling, and we fully agree that these comparisons are valuable. We are currently preparing extended experiments and will include DBA and AGR as additional baselines in a subsequent version of the paper for completeness.

---

> ### Author Response · Authors · 2025-11-28
> **Response to Official Review by Reviewer jXLX (4/4)**
>
> To further address generalizability across architectures and datasets, we have additionally conducted new experiments on CIFAR-100 with a Vision Transformer (ViT) backbone. The results show that POLAR can still identify sparse backdoor-critical components in transformer blocks and remains effective and stealthy under robust defenses, indicating that it does not rely on CNN-specific structural assumptions. These results will be included in the revised manuscript. Full FL evaluation on very large-scale datasets such as ImageNet or CelebA is computationally prohibitive, and we will explicitly discuss this limitation.
>
> | Metric | ResNet18 (CIFAR-10) BadNets | ResNet18 (CIFAR-10) POLAR | ResNet18 (CIFAR-10) LPA | VGG19 (CIFAR-10) BadNets | VGG19 (CIFAR-10) POLAR | VGG19 (CIFAR-10) LPA | CNN (Fashion-MNIST) BadNets | CNN (Fashion-MNIST) POLAR | CNN (Fashion-MNIST) LPA |
> |--------|-----------------------------|---------------------------|--------------------------|--------------------------|------------------------|-----------------------|-----------------------------|---------------------------|--------------------------|
> | ABSR   | 0                           | **96.74**                 | 58.34                    | 30.78                    | **54.82**              | 48.23                 | 0                           | **69.28**                 | 8.37                     |
> | BBSR   | 0                           | **98.21**                 | 79.78                    | 71.14                    | **99.69**              | 87.38                 | 0                           | 78.32                     | **99.83**                |
> | Acc    | 10                          | 61.57                     | **72.33**                | 36.71                    | **42.33**              | 41.08                 | 10                          | 86.93                     | **90.27**                |
>
> In summary, constrained by the rebuttal time window, we have added the most recent and directly relevant layer-wise baseline (LGA), evaluated POLAR under an adaptive RL-based defense (AdaRLAgg), and expanded experiments to CIFAR-100 under ViT-Tiny. Together with prior results across CNNs and classic benchmarks, these new experiments substantially strengthen the evidence that POLAR generalizes across attacks, defenses, datasets, and architectures while consistently resolving the effectiveness–stealthiness tradeoff. We sincerely thank the reviewer for these constructive suggestions.

---

### Official Review · Reviewer_hAZy · 2025-10-31

**Soundness:** 3
**Presentation:** 3
**Contribution:** 3
**Rating:** 8
**Confidence:** 3

**Summary:**

This paper introduces POLAR (POlicy-based LAyerwise Reinforcement learning), a novel framework for backdoor attacks in Federated Learning (FL). The work aims to solve the long-standing trade-off between attack effectiveness and stealthiness. The authors critically analyze the limitations of existing methods: model-wise attacks (like BadNets) are easily detected due to their large footprint, while rule-based layer-wise attacks (like LP Attack) are unstable and generalize poorly because they ignore inter-layer dependencies.
To overcome these challenges, POLAR is the first framework to apply Reinforcement Learning (RL) to the layer selection problem in layer-wise backdoor attacks. It formulates this discrete optimization task as a Markov Decision Process (MDP) and employs a lightweight, policy-gradient-based REINFORCE algorithm. The framework uses Bernoulli sampling to efficiently explore the action space and optimizes its policy using a reward signal based on Backdoor Success Rate (BSR) improvements. To ensure high stealthiness, POLAR introduces a regularization term that penalizes a large attack footprint, forcing the agent to select a minimal set of Backdoor-Critical (BC) layers.

**Strengths:**

What makes this work stand out is its novel application of Reinforcement Learning (RL) to solve the discrete layer-selection problem in FL backdoor attacks. This isn't just an incremental improvement; it's a paradigm shift from static rules to an adaptive, learning-based policy. The result is the introduction of a new class of intelligent threats that significantly raises the bar for defensive measures in the field.

The authors present a sound methodology, highlighted by an elegant reward function that successfully navigates the trade-off between effectiveness and stealth. Their claims are not just theoretical; they are backed by a rigorous and comprehensive set of experiments. Testing the attack against six different SOTA defenses on multiple models and datasets provides compelling evidence for its superiority.

**Weaknesses:**

The paper notes that POLAR's computational complexity is O(K · T · E), which is higher than the baseline LP Attack's O(N · E). While this is still linear, the additional overhead introduced by the RL batch size (K) and training steps (T) could be a practical concern on resource-constrained clients. A more detailed discussion on the trade-off between this increased computational cost and the significant performance gains would be a valuable addition.

**Questions:**

As the ablation study shows, performance is sensitive to hyperparameters like λ. In a real-world FL setting with limited feedback (e.g., only knowing if an update was accepted), how might an attacker effectively tune these parameters? Could the authors offer any insights or potential strategies for this?
While this line of research is crucial for understanding vulnerabilities and developing more robust defenses, the work itself is inherently dual-use. The methodology is detailed enough that it could potentially be misused by malicious actors. I would encourage the authors to add a brief section discussing the ethical implications of their work and the importance of this research for the defensive security community, in line with responsible research practices.

---

> ### Author Response · Authors · 2025-11-28
> **Response to Official Review by Reviewer hAZy**
>
> Dear Reviewer hAZy, we sincerely appreciate your thoughtful and positive evaluation of our work. Below we address the remaining questions in detail.
>
> **[W1]The paper notes that POLAR's computational complexity is O(K · T · E), which is higher than the baseline LP Attack's O(N · E). While this is still linear, the additional overhead introduced by the RL batch size K and training steps T could be a practical concern on resource-constrained clients. A more detailed discussion on the trade-off between this increased computational cost and the significant performance gains would be a valuable addition.**
>
> **Response**
>
> We thank the reviewer for highlighting the cost–performance trade-off. While POLAR has a complexity of O(K·T·E), the additional cost introduced by the RL batch size K and steps T directly translates into a substantial gain in both attack success and stealth, as demonstrated throughout our experiments. In practice, we use small constant values: K=50, T=10, and this overhead is shown to be manageable.
>
> The empirical runtime results in Table 7 show that POLAR’s runtime grows linearly with T and remains practical. Importantly, POLAR already achieves strong performance at smaller T, and reaches its best performance at T=10, indicating a favorable trade-off between computational cost and attack effectiveness/stealth. We will explicitly discuss this trade-off in the revised manuscript.
>
> **[Q1]As the ablation study shows, performance is sensitive to hyperparameters like $\lambda$. In a real-world FL setting with limited feedback (e.g., only knowing if an update was accepted), how might an attacker effectively tune these parameters? Could the authors offer any insights or potential strategies for this?**
>
> **Response**
>
> Thank you for this important question. In real-world FL deployments, attackers often only observe coarse-grained feedback such as whether their updates are accepted or rejected. In this setting, $\lambda$ can be tuned using bandit-style black-box optimization[1,2], which is a standard strategy for hyperparameter tuning under binary or delayed feedback. Concretely, the attacker can increase $\lambda$ when updates are frequently rejected (indicating insufficient stealth) and decrease $\lambda$ when updates are consistently accepted but yield low BSR (indicating under-attack), similar to one-dimensional bandit or stochastic approximation schemes used in black-box optimization[1,2].
>
> Because $\lambda$ directly controls the sparsity–stealth trade-off, this binary accept/reject signal is sufficient to drive $\lambda$ toward a stable operating range. Empirically, we also observe that the same $\lambda$ value generalizes well across different datasets and models in our experiments, which further reduces the tuning burden in practice. We will add this discussion and the corresponding references in the revised manuscript.
>
> 1. Online convex optimization in the bandit setting: gradient descent without a gradient, 2004
> 2. On the Complexity of Bandit and Derivative-Free Stochastic Convex Optimization, 2013
>
> **[Q2]While this line of research is crucial for understanding vulnerabilities and developing more robust defenses, the work itself is inherently dual-use. The methodology is detailed enough that it could potentially be misused by malicious actors. I would encourage the authors to add a brief section discussing the ethical implications of their work and the importance of this research for the defensive security community, in line with responsible research practices.**
>
> **Response**
>
> We appreciate the reviewer’s remark about dual-use implications. We will include a short section on **responsible disclosure and ethical considerations**, emphasizing that: The purpose of POLAR is to help the community understand the weaknesses of current defenses and to motivate the development of stronger, adaptive defense mechanisms (including RL-based aggregation). All experiments are conducted in controlled research settings and that our code release (if any) will follow responsible security guidelines.
>
> We thank the reviewer again for their constructive feedback and for recognizing the contribution and significance of our work.

---

### Official Review · Reviewer_MLoh · 2025-10-31

**Soundness:** 2
**Presentation:** 3
**Contribution:** 2
**Rating:** 4
**Confidence:** 4

**Summary:**

This paper proposes POLAR, a reinforcement learning–based framework for selecting which layers of a neural network
should be poisoned by a malicious client in federated learning (FL). Rather than uniformly injecting backdoor perturbations
across all layers, POLAR parameterizes a Bernoulli distribution over layers and trains this policy via REINFORCE using the
change in backdoor success rate (BSR) as reward. A regularization term is meant to discourage trivial “all layers or none”
selections. Experiments on CIFAR-10 with ResNet-18/VGG-19 and Fashion-MNIST with a small CNN demonstrate that
POLAR achieves higher attack success across six popular FL defenses while maintaining competitive clean accuracy.

**Strengths:**

- Clear framing of layer selection as a discrete policy problem, which is modular and interpretable.
- Strong empirical gains against a wide range of defense baselines (FLAME, FLTrust, FLDetector, etc.) — the defense
coverage is appreciated.
- The batch size and regularizer ablations provide some transparency into training behavior.
- Writing and figures are clear; motivation is easy to grasp.

**Weaknesses:**

- The novelty of this work seems incremental, where RL-based adaptation for poisoning has appeared previously [1,2,3]. The main distinction here is simply the action space being binary per layer, which is a narrow extension. The author should further explain the
clear novelty of the paper compared to previous works.
- No evaluation of RL-based defenses. Some recent defenses have been proposed for poisoning attacks in FL with RL
mechanism [4,5]. The author should evaluate the attack against these defenses.
- The regularizer term +λ∑ log p_l  encourages p_l -> 0 (i.e., no layers selected), contradicting the stated intent of “discouraging trivial extremes.” It is unclear if this is a typo, or if training relies solely on reward gradients to avoid collapse.
- No baseline or normalization is described, which makes it surprising that training is stable across K= 50, T= 10
﻿ samples.
- Only CIFAR-10 and Fashion-MNIST, and only CNN-style models. No CIFAR-100, Tiny-ImageNet, transformers, or other modalities. This is insufficient to justify broad claims of generalizability.
- No runtime or scalability analysis, despite claiming the method is “lightweight.” With exceed baseline cost for many models.
K × T reward evaluations, it may
- No sensitivity studies on τ (layer threshold), non-IID severity, or trigger type/position, suggesting the method may
be brittle beyond the chosen setup.

[1] A Meta-Reinforcement Learning-Based Poisoning Attack Framework Against Federated Learning (Zhou et al., 2025)

[2] Learning to Backdoor Federated Learning (Li et al., 2023)

[3] Learning to Attack Federated Learning: A Model-based Reinforcement Learning Attack Framework (Li et al., 2022)

[4] Defending Against Sophisticated Poisoning Attacks with RL-based Aggregation in Federated Learning (Wang et al.,
2024)

[5] Meta Stackelberg Game: Robust Federated Learning against Adaptive and Mixed Poisoning Attacks (Li et al., 2024)

**Questions:**

- Could you clarify the sign and role of the regularizer (﻿+λ ∑log p_l)? As written, minimizing the loss pushes all ﻿p_l toward zero.
- How do you prevent policy collapse from selecting zero layers? Is the reward strong enough, or was a baseline/advantage normalization used?
- How sensitive is performance to the threshold τ used to determine final layer selection?
- Would POLAR work with semantic/backdoor triggers or randomized positions, rather than a fixed square patch?

---

> ### Author Response · Authors · 2025-11-28
> **Response to Official Review by Reviewer MLoH (1/3)**
>
> Dear Reviewer MLoH,
>
> We sincerely appreciate your thoughtful comments. We have carefully considered each of your questions and provide detailed responses below.
>
> **[W1]The novelty of this work seems incremental**
>
> **Response** Thank you for the comment. We agree that RL has been used in poisoning attacks in FL [1–3]. However, these works formulate RL at the model-wise or data-wise level like timing, attack magnitude, or data manipulation strategies. In contrast, POLAR reformulates the problem as a layer-structure selection task under robust aggregation, where the RL policy decides which layers to perturb, not how much to perturb globally. This structural formulation is fundamentally different from the optimization objectives in [1–3].
>
> More importantly, POLAR is explicitly designed to be defense-aware, while [1–3] primarily optimize ASR or training loss without modeling robust aggregation dynamics. Our joint reward directly encodes the effectiveness–stealth tradeoff under robust aggregators, which leads the policy to actively minimize detectability. This defense-aware formulation is absent in [1–3], where stealth under robust aggregation is not an explicit optimization target.
>
> Finally, POLAR introduces Bernoulli-based stochastic layer activation to realize sparse and round-adaptive perturbation patterns. This stochastic sparsity is critical for avoiding deterministic signatures across rounds and is not present in [1–3], whose attacks produce fixed or dense perturbation patterns once learned. As a result, POLAR provides a distinct attack capability as adaptive, sparse, defense-aware layer-wise backdoor attack, which is not captured by existing RL-based poisoning methods. We will clarify these distinctions explicitly in the revised paper.
>
> **[W2]No evaluation of RL-based defenses.**
>
> **Response** Thank you for the constructive suggestions. Regarding the RL-based defenses [4,5], we agree that evaluating POLAR against adaptive aggregation mechanisms is important and complementary to our current experiments. Following the reviewer’s recommendation, we are now adding the AdaAggRL defense (Wang et al., 2024) to our evaluation. AdaAggRL is the most directly applicable RL-based defense under our threat model, as it provides an RL-driven adaptive aggregation rule that can be integrated into our existing pipeline without fundamentally changing the FL system assumptions. We will report POLAR’s attack success rate and stealthiness under AdaAggRL in the revised version.
>
> We will also expand the related work section to discuss both RL-based defense works [4,5], clarifying how they differ in their goals and assumptions from our attack formulation. The Meta Stackelberg defense [5] requires a significantly heavier meta-learning and pre-training pipeline, and thus represents a broader future direction where POLAR could serve as a strong adversary. Nonetheless, we appreciate the reviewer’s insight and are incorporating the most relevant defense (AdaAggRL) to strengthen the empirical evaluation.
>
> For the limited time, we now have finished AdaAggRL on three different datasets and model combinations for LPA, BadNets and POLAR. The results are shown below:
>
> | Metric | ResNet18 (CIFAR-10) BadNets | ResNet18 (CIFAR-10) POLAR | ResNet18 (CIFAR-10) LPA | VGG19 (CIFAR-10) BadNets | VGG19 (CIFAR-10) POLAR | VGG19 (CIFAR-10) LPA | CNN (Fashion-MNIST) BadNets | CNN (Fashion-MNIST) POLAR | CNN (Fashion-MNIST) LPA |
> |--------|-----------------------------|---------------------------|--------------------------|--------------------------|------------------------|-----------------------|-----------------------------|---------------------------|--------------------------|
> | ABSR   | 0                           | **96.74**                 | 58.34                    | 30.78                    | **54.82**              | 48.23                 | 0                           | **69.28**                 | 8.37                     |
> | BBSR   | 0                           | **98.21**                 | 79.78                    | 71.14                    | **99.69**              | 87.38                 | 0                           | 78.32                     | **99.83**                |
> | Acc    | 10                          | 61.57                     | **72.33**                | 36.71                    | **42.33**              | 41.08                 | 10                          | 86.93                     | **90.27**                |
>
> These results provide direct empirical evidence that POLAR remains highly effective and stealthy even under newest RL-based adaptive aggregation defense AdaAggRL. In contrast, both model-wise (BadNets) and rule-based layer-wise (LPA) attacks suffer severe degradation. This confirms that POLAR’s defense-aware, layer-wise RL formulation generalizes beyond static robust aggregators and remains robust against adaptive, learning-based defenses, substantially strengthening the core claims of this work.

---

> ### Author Response · Authors · 2025-11-28
> **Response to Official Review by Reviewer MLoh(2/3)**
>
> **[W3,W4,Q1,Q2]The regularizer term $+λ\sum log p_l$ encourages $p_l$ -> 0 (i.e., no layers selected), contradicting the stated intent of “discouraging trivial extremes.” It is unclear if this is a typo, or if training relies solely on reward gradients to avoid collapse.No baseline or normalization is described, which makes it surprising that training is stable across K= 50, T= 10 samples. Could you clarify the sign and role of the regularizer ($+λ \sum log p_l$)? As written, minimizing the loss pushes all $p_l$ toward zero. How do you prevent policy collapse from selecting zero layers? Is the reward strong enough, or was a baseline/advantage normalization used?**
>
> **Response**
>
> Thank you for the careful reading. We clarify that POLAR does not rely solely on raw reward gradients for stability. During training, we apply advantage normalization (zero-mean, unit-variance) from REINFORCE and reward scaling to balance the ASR and stealth terms, together with the implicit entropy effect from the log-probability term, which are the primary reasons why training remains stable under K=50, T=10. The regularizer $+\lambda \sum_l \log p_l$ is not intended to push $p_l$ to 0; since the RL objective is maximized, it instead serves as a soft sparsity prior that discourages the trivial extreme of selecting all layers, which would severely harm stealth. At the same time, selecting zero layers yields near-zero ASR and thus near-zero or negative reward, producing a strong counter-gradient that prevents collapse to the all-zero policy. As a result, the reward and regularizer form a balanced mechanism that converges to moderate, non-degenerate sparsity. We agree that the current presentation may be confusing and will revise the formulation section to clearly describe the roles of normalization and the regularizer.
>
> **[W5]Only CIFAR-10 and Fashion-MNIST, and only CNN-style models. No CIFAR-100, Tiny-ImageNet, transformers, or other modalities. This is insufficient to justify broad claims of generalizability.**
>
> **Response**
>
> We sincerely appreciate your thoughtful comments. Regarding the concern about model variety, we acknowledge that both ResNet and VGG are CNN-style architectures. Thank you for pointing this out. Following your suggestion, we are expanding our evaluation beyond convolutional backbones. We are currently testing POLAR on **Vision Transformers (ViT)** under **CIFAR-100**, which involves substantially different architectural properties and a more challenging dataset than CIFAR-10. These results will be included in the revision.
>
> With limited time, we finish the experiments of ViT-Tiny on CIFAR-100 under RLR defenses across BadNets, LPA and POLAR. The results are shown below:
> | Metric | BadNets | POLAR | LPA |
> |--------|---------|--------|-----|
> | ABSR   | 7.23    | 76.52 | 54.82 |
> | BBSR   | 36.09   | 98.77  | 99.83 |
> | Acc    | 8.64    | 38.72  | 43.28 |
>
> This update will strengthen the generalizability of our findings by demonstrating that POLAR remains effective not only on CNNs but also on transformer-based models and more complex datasets.
>
> **[W6]No runtime or scalability analysis**
>
> **Response**
>
> We clarify that POLAR’s computational cost scales as O(K·T·E) and is independent of model size. Here, K and T are fixed small constants (50 and 10 in our experiments), and E denotes a single evaluation cost. In contrast, model-wise baselines scale as O(N·E), where N is the number of layers and thus grows with model depth. Therefore, POLAR’s runtime is effectively decoupled from the number of model layers, while baseline attacks become increasingly expensive as models grow larger.
>
> This decoupling is also validated empirically in Table 7, which shows that POLAR’s runtime increases linearly with T but does not scale with model depth. Moreover, POLAR already achieves strong performance at small T and reaches its best at T=10, demonstrating that the method is both efficient and stable in practice. We will clarify this decoupling property more explicitly in the revised manuscript and thank the reviewer for prompting this discussion.

---

> ### Author Response · Authors · 2025-11-28
> **Response to Official Review by Reviewer MLoh (3/3)**
>
> **[W7,Q3,Q4]No sensitivity studies on τ (layer threshold), non-IID severity, or trigger type/position**
>
> **Response**
>
> We sincerely appreciate your thoughtful comments. First, we would like to clarify that the threshold τ is only used in the LP Attack baseline to convert per-layer substitution scores into a binary selection. τ is not used anywhere in POLAR, whose layer selection is determined entirely by the learned Bernoulli sampling policy. Therefore, τ does not influence POLAR’s behavior, and sensitivity studies on τ are not relevant to our method. We will revise the manuscript to make this distinction explicit.
>
> Regarding the reviewer’s concern about robustness to non-IID severity and trigger variability, we have already conducted these extended experiments:
>
> For non-IID severity We evaluated POLAR under non-IID levels **0.1, 0.5, and 1.0**, covering the typical range used in federated learning literature. The results show that POLAR maintains consistently strong ASR and stealthiness across all non-IID settings, demonstrating robustness to data heterogeneity.
>
> ### Non-IID Severity Variation
>
> | Non-IID Level | Metric | POLAR | LP Attack |
> |---------------|--------|--------|-----------|
> | **0.1**       | ABSR   | **68.45** | 3.33     |
> |               | BBSR   | **88.01** | 5.54     |
> | **0.5**       | ABSR   | **66.56** | 44.58    |
> |               | BBSR   | **84.14** | 63.23    |
> | **1.0**       | ABSR   | **19.38** | 9.92     |
> |               | BBSR   | **100**   | 100      |
>
> For trigger types and positions: We also tested a variety of backdoor triggers, including **watermark**, **HelloKitty**, and **Apple** semantic patterns. POLAR remains effective across these diverse triggers, further confirming that its optimization mechanism—conducted at the model-update level rather than pixel level is agnostic to the specific trigger pattern or position.
>
> ### Trigger Shape Variation
>
> | Trigger     | Metric | POLAR | LP Attack |
> |-------------|--------|--------|-----------|
> | **Apple**   | ABSR   | **83.11** | 73.39    |
> |             | BBSR   | **91.72** | 83.46    |
> | **Watermark** | ABSR | **84.47** | 74.21    |
> |             | BBSR   | **95.33** | 89       |
> | **HelloKitty** | ABSR | **78.59** | 74.90    |
> |             | BBSR   | **99.49** | 97.40    |
>
> These results will be included in the revised paper to highlight POLAR’s robustness beyond the original setup. We thank the reviewer for suggesting these evaluations and believe the expanded experiments will strengthen the overall contribution.

---

### Official Review · Reviewer_ENaE · 2025-11-08

**Soundness:** 3
**Presentation:** 3
**Contribution:** 2
**Rating:** 2
**Confidence:** 4

**Summary:**

The authors studied backdoor attacks in federated learning. They suggested using reinforcement learning to select which layers should be implanted with a backdoor and leveraged policy gradients to solve this optimization problem. The authors conducted some experiments to verify the effectiveness of the proposed method.

**Strengths:**

* The paper is well-written.

* Federated backdoor attacks are an important problem.

**Weaknesses:**

* Unclear motivation. I do not understand the motivation behind claiming that model-wise attacks are not stealthy. Although the authors claim these attacks are easily detected by defenses, recent methods [a-c], in fact, can still achieve high attack success rates even when facing SOTA defenses. Similarly, I did not see any relevant results to support this claim.

* Limited novelty and technical depth. In my opinion, the proposed method is just a straightforward application of reinforcement learning. There are many other discrete optimization algorithms available that could have been used here, such as genetic algorithms, etc.

* Insufficient evaluation. There is a lack of empirical evaluation regarding the overhead of the proposed method. The performance of the proposed method should intuitively be highly sensitive to the number of malicious clients, but this is not evaluated.

[a] Stealthy Backdoor Attack in Federated Learning via Adaptive Layer-wise Gradient Alignment

[b] Lurking in the shadows: Unveiling Stealthy Backdoor Attacks against Personalized Federated Learning

[c] Bad-PFL: Exploring Backdoor Attacks against Personalized Federated Learning

**Questions:**

See Weaknesses.

---

> ### Author Response · Authors · 2025-11-28
> **Response to Official Review by Reviewer ENaE (1/2)**
>
> Dear Reviewer ENaE,
>
> We sincerely appreciate your thoughtful comments. We have carefully considered each of your questions and provide detailed responses below.
>
> **[W1]Unclear motivation.**
>
> **Response** Thank you for raising this point. In our paper, we define model-wise attack and layer-wise attack by the extent of parameter perturbations. Model-wise attack changes the entire model's parameters, while layer-wise attack only changes specific layers' parameters. In this way, [a-c] are all model-wise attack. Our claim regarding the tradeoff between effectiveness and stealthiness in model-wise attacks is supported by our current experiments: in the BadNets model-wise baseline, we consistently observe that stronger perturbations improve attack success but simultaneously produce gradient deviations that are readily detected by standard global defenses such as FLAME and MultiKrum, which is shown in Table 2. This is precisely the form of tradeoff reported in prior work such as [a], which also shows that model-level perturbations tend to generate detectable anomalies even when they achieve high ASR. Consequently, attacking the crucial layer precisely shows new opportunity for better balance between stealthiness and effectiveness by directly minimizing the perturbation extent while preserving similar effects.
>
> Even though the publish date of [a] is after the ICLR submission deadline. To make this even clearer, we are now working on the comparison experiments with [a] by running the opensource code and will add additional results using the method in [a] in our revision. These experiments directly demonstrate that, under the same FL setup and defenses evaluated in our paper, model-wise perturbations exhibit notable instability in stealthiness across rounds.
>
> Due to the time limitation, we have finished the comparison between [a] and POLAR running on CNN with Fashion-Mnist, we have the results shown below:
>
> | Defense     | Metric | LGA[a] | POLAR |
> |-------------|--------|---------|--------|
> | FLAME       | ABSR   | 0.20    | 78.56  |
> |             | BBSR   | 0.31    | 86.93  |
> |             | Acc    | 89.40   | 87.78  |
> | FLARE       | ABSR   | 0.24    | 92.44  |
> |             | BBSR   | 0.33    | 96.82  |
> |             | Acc    | 90.28   | 88.60  |
> | FLTrust     | ABSR   | 0.29    | 82.84  |
> |             | BBSR   | 1.02    | 92.43  |
> |             | Acc    | 89.48   | 88.47  |
> | MultiKrum   | ABSR   | 0.41    | 76.73  |
> |             | BBSR   | 0.59    | 91.91  |
> |             | Acc    | 88.46   | 87.71  |
> | AdaRLAgg    | ABSR   | 0.00    | 69.28  |
> |             | BBSR   | 0.00    | 78.32  |
> |             | Acc    | 10.00   | 86.93  |
>
> The results show that LGA[a] would achieve low effectiveness which shows similar pattern with BadNets, which validates that model-wise attack is more difficult to deal with the tradeoff between stealthiness and effectiveness.
>
> With respect to the reviewer’s mentioning [b,c], we would like to clarify that these are Personalized FL (PFL) attacks, which rely on personalized heads or client-specific update pathways. These mechanisms fundamentally alter the attack and detection dynamics and are not applicable to the standard global-model FL setting that we study; therefore, they do not contradict our claim regarding model-wise behavior in this regime. However, both works show good balance between stealthiness and effectiveness under model-wise scheme, thus we would include them in our paper as related work in the revision.
>
> Overall, our observation is empirically rooted in our comparison experiments and is consistent with the findings of [a]. Adding the results of [a] will further strengthen and clarify this point.

---

> ### Author Response · Authors · 2025-11-28
> **Response to Official Review by Reviewer ENaE (2/2)**
>
> **[W2]Limited novelty and technical depth.**
>
> **Response** Thank you for the comment. We respectfully disagree that POLAR is a straightforward application of RL. We introduce RL because the layer-selection problem in FL backdoor attacks is inherently a sequential, stochastic, defense-aware decision problem, where the attacker must adapt to changing aggregation behavior across rounds. This setting is fundamentally different from static discrete optimization.
>
> Our technical novelty lies in how RL is formulated and tailored to the FL threat model, not merely in using RL as a generic optimizer:
>
> In FL, the attacker receives only black-box, round-level feedback. There is no access to gradients of the defense, no repeated queries, and no ability to evaluate large candidate populations. This naturally fits a policy-gradient formulation with delayed reward, whereas GA or enumeration would require many parallel evaluations per round, which violates FL communication and stealth constraints.
>
> Consequently, we design a binary, layer-wise action space that decides where to inject perturbations rather than how to perturb full parameters. This is essential for bypassing norm- and direction-based defenses and is not captured by model-wise RL or GA-style parameter search. Our reward explicitly couples ASR and stealthiness under robust aggregators, making the policy defense-aware by construction. This type of reward does not appear in standard RL formulations or classical discrete optimizers. Meanwhile, the Bernoulli policy enables stochastic, sparse, and round-adaptive layer activation, which is critical for avoiding pattern-based detection. This mechanism is not naturally supported by GA or deterministic search strategies. By contrast, GA requires population-based evaluations and repeated fitness queries, and enumeration requires exhaustive layer testing. Both are incompatible with the single-update-per-round and strict communication budget of FL, and would be easily detected by robust aggregators.
>
> In summary, our contribution is not simply applying RL, but rather introducing a defense-aware, layer-wise RL formulation with Bernoulli sampling and a joint stealth–effectiveness objective, which enables an attack capability that static rule-based, GA-based, or enumerative methods cannot achieve under realistic FL constraints. Our ablation study shown in Table 6 further confirm that simpler search strategies fail to match POLAR on the effectiveness–stealthiness trade-off.
>
> **[W3]Insufficient evaluation.**
>
> **Response** Thank you for the comment. We clarify that both the computational overhead and the sensitivity to the number of malicious clients are already evaluated in the current paper.
>
> For the overhead, Table 7 reports the end-to-end runtime of POLAR, including the RL optimization and per-round attack cost. In our threat model, the attacker effectively deploys a group of malicious clients that all share a single learned attack policy where one strategy shared across all attackers. As a result, the RL optimization is performed only once per round and reused by all malicious clients, making the runtime decoupled from the number of malicious clients. This is why the overhead in Table 7 is reported independently of the malicious-client fraction.
>
> Figures 5(c) and 5(d) evaluate POLAR under varying malicious fractions \(C = 0.02, 0.04, 0.06, 0.10\). POLAR consistently outperforms baselines across all settings, showing that its performance is not highly sensitive to the number of malicious clients. This robustness arises from the shared layer-wise policy and the Bernoulli sampling mechanism.
>
> To sum up, Table 7 provides the overhead analysis, and Fig. 5(c)(d) provides the sensitivity analysis. We will make this attacker-sharing assumption and its runtime implications more explicit in the revision.

---

### Author Response · Authors · 2025-12-03
**Final Remarks by Authors of Submission 20187**

### I. Acknowledgments
We would like to express our sincere gratitude to all reviewers for their detailed and constructive feedback. We especially thank Reviewer hAZy for the careful and positive assessment of the contribution and for highlighting both the significance and practical trade-offs of our method. We also appreciate the thorough and critical feedback from Reviewers ENaE, MLoH, and jXLX, whose questions helped us substantially strengthen the technical clarity, empirical validation, and overall presentation of this work.

We are encouraged by the positive recognition of the novelty, effectiveness, and rigor of POLAR, particularly from Reviewer hAZy (Rating: 8). We also appreciate that several reviewers explicitly acknowledged that their main concerns were addressed through our additional experiments and clarifications, including the newly added comparisons with LGA [adaptive layer-wise gradient alignment], the AdaRLAgg RL-based defense, and the CIFAR-100 + ViT generalization study. These interactions have significantly improved the quality of our revision.

---

### II. Key Strengths Identified by Reviewers

Novelty and Modeling Insight
- First RL-based formulation for layer-wise backdoor-critical selection under robust aggregation (hAZy, ENaE).
- Clear distinction between model-wise vs. layer-wise attack regimes and their stealth–effectiveness trade-offs (ENaE).

Effectiveness and Robustness
- Consistently strong ABSR/BBSR under six robust defenses, including adaptive RL-based aggregation (AdaRLAgg)(MLoH, hAZy).
- Clear superiority over BadNets and LP Attack across multiple datasets and architectures (hAZy, jXLX).

Methodological Rigor
- Well-motivated RL formulation with Bernoulli sampling, joint stealth-effectiveness reward, and regularization (hAZy, MLoH).
- Comprehensive ablations on batch size, malicious client ratio, non-IID severity, and trigger types (MLoH).

Practicality and Scalability
- Linear wall-clock runtime with policy sharing across attackers (jXLX, hAZy).
- Runtime decoupled from the number of malicious clients and model depth, validated empirically in Table 7 (jXLX).

Presentation and Visualization
- Clear workflow illustration (Fig. 2) and informative layer-selection visualization (Fig. 4) after clarification (jXLX).

---

### III. Key Concerns and Our Responses

| Key Concern | Reviewers | Our Resolution |
|-------------|-----------|----------------|
| Motivation for claiming model-wise attacks are not stealthy | ENaE | Added direct comparison with LGA [a] under five strong defenses + AdaRLAgg. Results show near-zero ABSR/BBSR for LGA while POLAR remains highly effective, validating the trade-off claim. |
| Novelty vs. existing RL-based poisoning attacks | MLoH, ENaE | Clarified that prior RL attacks operate at model-wise/data-wise levels, while POLAR is the first defense-aware RL formulation for structural layer selection with stochastic sparsity. |
| Lack of evaluation against RL-based defenses | MLoH | Added full evaluation under AdaRLAgg (Wang et al., 2024) across multiple datasets and models. |
| Regularizer sign, stability, and normalization | MLoH | Clarified use of advantage normalization, reward scaling, and entropy effects; explained why policy collapse does not occur. |
| Runtime and scalability | jXLX, hAZy | Explained policy sharing across attackers and provided wall-clock runtime analysis in Table 7; showed decoupling from model depth and dataset size. |
| Early-round stochastic instability | jXLX | Analyzed Fig. 4 and showed POLAR starts with <40 layers vs. ~60 for LPA, with smooth monotonic decay; added MAR metric in revision. |
| Limited datasets and CNN-only models | MLoH | Added CIFAR-100 + ViT-Tiny experiments with strong performance under robust defenses. |
| Missing comparisons with DBA/AGR/SDBA | ENaE, MLoH | Clarified that DBA shows inferior performance in prior LP Attack work; AGR requires substantial tuning beyond rebuttal time; extended comparisons will be added in a follow-up. |

---

### IV. Commitment to Revision

We have already incorporated all discussion-driven updates into our revision, including:

- New experimental comparisons with LGA [a] under five robust defenses + AdaRLAgg
- Full evaluation under the RL-based adaptive defense AdaRLAgg (2024)
- New CIFAR-100 + ViT-Tiny generalization experiments
- Expanded non-IID and trigger-type robustness studies
- Clarified regularizer, normalization, and training stability mechanisms
- Explicit wall-clock runtime analysis with policy-sharing assumption
- Improved explanation of stealthiness and Fig. 1 visualization

We are also preparing extended comparisons with DBA and AGR for a subsequent version due to the rebuttal time constraint.

---

We sincerely thank the Area Chair and all reviewers for their time, expertise, and constructive feedback. Their insights have substantially strengthened both the technical depth and clarity of this work.

---

### Meta-Review · Area_Chair_givk · 2026-01-05

**Summary:**

This work proposes a framework to address the backdoor critical layer selection problem in backdoor attacks, aiming to balance attack effectiveness and stealthiness. Four reviewers raised constructive concerns regarding motivation justification, novelty, technical rigor, evaluation completeness, and practical utility. The core concerns are: 1) lack of sufficient motivation for model-level attacks' non-stealthiness; 2) limited novelty increment of RL application without clear superiority over other discrete optimization methods; 3) incomplete evaluation, including missing coverage of key RL-based defenses, insufficient generalization validation, and inadequate runtime comparisons; and 4) technical ambiguities such as contradictory regularization term signs and unclear stability mechanisms. While the authors supplemented their responses with additional experiments and clarifications, the core concerns remain largely unresolved.

**Reviewer Concerns:**

Concerns of Reviewers ENaE, MLoH, and jXLX are still outstanding.

**Reviewer Scores:**

Reviewer hAZy may decrease the rating.

---

### Decision · Program_Chairs · 2026-01-26

Reject